# Structural basis for gating mechanism of the human sodium-potassium pump

Phong T. Nguyen[1] ✉, Christine Deisl [2], Michael Fine [2], Trevor S. Tippetts [3], Emiko Uchikawa[4], Xiao-chen Bai [4] ✉ & Beth Levine[1,5]

P2-type ATPase sodium-potassium pumps (Na+/K+-ATPases) are ion-transporting enzymes that use ATP to transport Na+ and K+ on opposite sides of the lipid bilayer against their electrochemical gradients to maintain ion concentration gradients across the membranes in all animal cells. Despite the available molecular architecture of the Na+/K+-ATPases, a complete molecular mechanism by which the Na+ and K+ ions access into and are released from the pump remains unknown. Here we report five cryo-electron microscopy (cryo-EM) structures of the human alpha3 Na+/K+-ATPase in its cytoplasmic side-open (E1), ATP-bound cytoplasmic side-open (E1•ATP), ADP-AlF₄⁻ trapped Na+-occluded (E1•P-ADP), BeF₃⁻ trapped exoplasmic side-open (E2P) and MgF₄²⁻ trapped K+-occluded (E2•Pᵢ) states. Our work reveals the atomically resolved structural detail of the cytoplasmic gating mechanism of the Na+/K+-ATPase.

The type-2 P-type sodium-potassium pump (Na+/K+-ATPase), first discovered in the 1950's by Jens Christian Skou[1], plays significant roles in maintaining the electrochemical gradients for Na+ and K+ across the plasma membrane of animal cells. By utilizing energy from the hydrolysis of a single ATP, the Na+/K+-ATPase pumps three Na+ ions out and two K+ ions into the cell in each transport cycle. Structurally, the Na+/K+-ATPase is composed of three subunits assembled in a 1:1:1 stoichiometry: (1) alpha (α)– a catalytic unit (~110 kDa) that consists of three cytoplasmic domains (A, actuator; N, nucleotide binding; and P, phosphorylation) and a 10-transmembrane (TM)-helix domain (TMD); (2) beta (β)– a single TM extracellular unit (~35 kDa) required for Na+/K+-ATPase's stability and membrane trafficking; (3) FXYD—a single TM regulatory unit (~10 kDa) that modulates the enzymatic activity by changing affinities of the Na+/K+-ATPase toward Na+, K+ or ATP (Fig. 1a)[2,3]. In humans, four α, three β, and seven FXYD isoforms have been identified, among which the expression of the α isoforms is tissue-specific. The α1 subunit is ubiquitously expressed in all tissues, whereas the α2, α3, and α4 are predominantly detected in muscle, brain and testis, respectively[3,4]. Currently, the only isoform structure determined by X-ray crystallography is the ubiquitously expressed α1 configuration[5–11].

It is proposed that the transport mechanism of the Na+/K+-ATPase follows the Post−Albers scheme−cycling between the E1 and E2 states. These states associate with Na+-dependent autophosphorylation and K+-dependent dephosphorylation, respectively. In the E1 state, the Na+/K+-ATPase adopts a cytoplasmic side-open conformation where the ion-binding sites open to the cytoplasm, allowing Na+ to access the transmembrane binding sites. Upon Mg-ATP and 3 Na+ binding, the Na+/K+-ATPase undergoes autophosphorylation at the conserved aspartate residue in the P domain of the α subunit, thereby switching to an Na+-occluded E1•P-ADP state where the ion-binding sites are not accessible to the cytoplasm. Upon ADP release, the Na+/K+-ATPase switches to E2P state and the ion-binding sites open to the exoplasmic side for Na+ release and subsequent 2 K+ binding. The binding of K+ triggers dephosphorylation and drives the conformational change of the Na+/K+-ATPase to an K+-occluded state (E2•Pi). The Na+/K+-ATPase subsequently switches to the E2 state, and finally back to an E1 state releasing K+ into the cytoplasm allowing for the transport cycle to begin anew (Fig. 1b)[2,12].

Previous X-ray crystallographic work revealed the architecture of the Na+/K+-ATPase in two distinct Na+-occluded (E1•P-ADP)[7,10] and K+-occluded (E2•Pi)[5,9] states. Recent reports of the exoplasmic side-open

[1]Howard Hughes Medical Institute and Department of Internal Medicine, University of Texas Southwestern Medical Center, Dallas, TX, USA. [2]Department of Physiology, University of Texas Southwestern Medical Center, Dallas, TX, USA. [3]Children's Research Institute, University of Texas Southwestern Medical Center, Dallas, TX, USA. [4]Department of Biophysics, University of Texas Southwestern Medical Center, Dallas, TX, USA. [5]Deceased: Beth Levine. ✉e-mail: phong.nguyen@utsouthwestern.edu; xiaochen.bai@utsouthwestern.edu

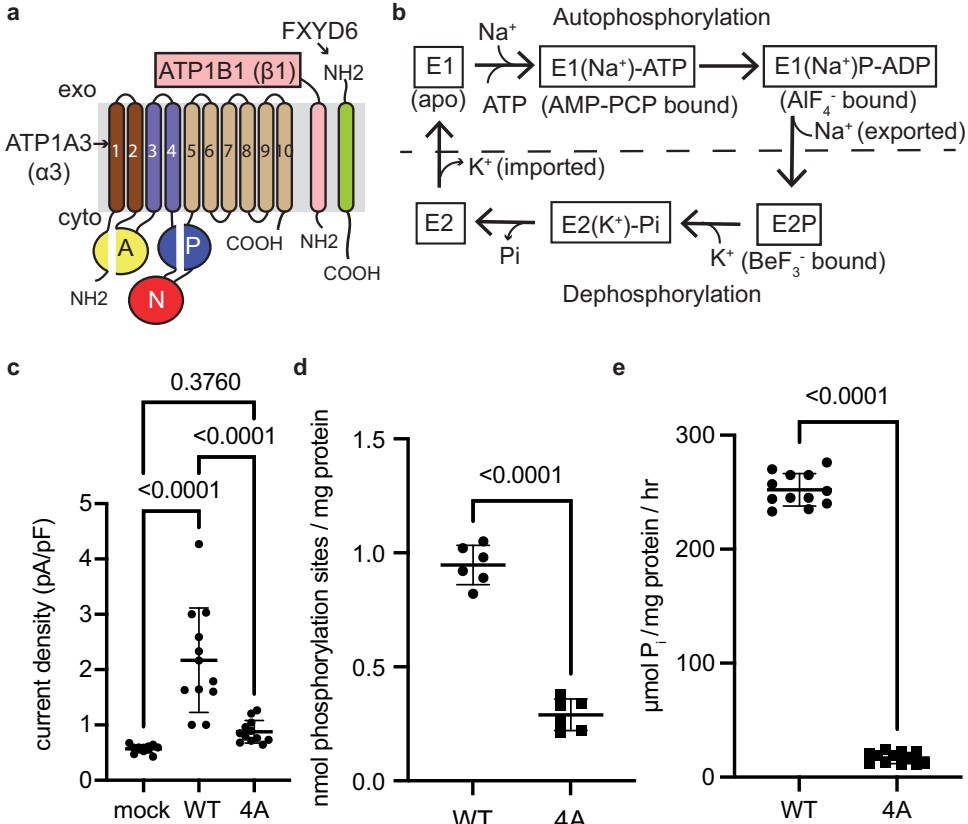

**Fig. 1 | Functional characterization of the human α3 Na⁺/K⁺-ATPase. a** Topology of the Na⁺/K⁺-ATPase. **b** The Post–Albers model for the Na⁺/K⁺-ATPase. **c** Whole-cell patch-clamp electrophysiological current density (pA/pF) of the wild-type (WT) and cation-binding deficient (4A) mutant Na⁺/K⁺-ATPase. $N = 12$ cells examined over four independent experiments. One-way ANOVA test was used to compare mean values of different variants. *P*-values were shown as numerical numbers in the subfigure 1c. Source data are provided as a Source Data file. **d**, **e** The numbers of phosphorylation sites (**d**) ($n = 6$ independent experiments) and ATP turnover rates (**e**) ($n = 12$ independent experiments) of the wild-type (WT) and cation-binding deficient (4A) mutant Na⁺/K⁺-ATPase. Unpaired T-test was used to compare mean values of different variants. Data were presented as mean ± standard deviation (SD). *P*-values were shown as numerical numbers in the subfigure 1d and e. Source data are provided as a Source Data file.

(E2P) structure of the Na⁺/K⁺-ATPase indicates how the exoplasmic-gating mechanism works to release bound Na⁺ and facilitate extracellular K⁺ access into the pump's ion-binding cavities[11,13]. However, the cytoplasmic gating mechanism of the Na⁺/K⁺-ATPase still remains unclear due to the lack of structural information of the cytoplasmic side-open states (E1 and E1·ATP). Here, we determined the cryo-EM structures of the human Na⁺/K⁺-ATPase in five intermediate states during its transport cycle, including the cytoplasmic side-open (E1), AMPPCP-bound cytoplasmic side-open (E1·ATP), ADP-AlF₄⁻ trapped Na⁺-occluded (E1·P-ADP), BeF₃⁻ trapped exoplasmic side-open (E2P) and MgF₄²⁻ trapped K⁺-occluded (E2·Pᵢ) states. Structural comparison among these five states elucidated the conformational rearrangements in both cytoplasmic and transmembrane domains that are critical for the gating mechanism of the Na⁺/K⁺-ATPase. In brief, the opening cytoplasmic gate for Na⁺ access is triggered by a hinge-type rotation of the TMD's M1 helix distal to M3 helix, while opening of the exoplasmic gate for K⁺ access is triggered by a twist of M4E away from M6. These structural insights and functional validation assays reveal structural mechanistic details of Na⁺/K⁺-ATPase gating.

## Results

### Construct optimization and functional characterization of the recombinant human α3-isoform Na⁺/K⁺-ATPase

To exogenously reconstitute the human α3 Na⁺/K⁺-ATPase, we screened different combinations of βs and FXYDs with α3, and found out that a combination of the α3, β1 and FXYD6 yielded the highest expression level. The three genes encoding for the α3, β1 and FXYD6 were cloned into a single vector containing the CMV promoter and the P2A ribosomal skipping sequences in between the three genes (see Methods for more details). The baculovirus-mammalian (bacmam) expression system was utilized to express the human α3 Na⁺/K⁺-ATPase. The Flag-tagged α3 Na⁺/K⁺-ATPase was purified by anti-Flag antibody-conjugated resin (FlagM2) (Sigma). The purified Na⁺/K⁺-ATPase was shown as a single peak on gel filtration (Supplementary Fig. 1a) and fully assembled with the three subunits (α3, β1 and FXYD6) (> 95% purity) as shown by SDS-PAGE analysis (Supplementary Fig. 1b). To validate the functionality of this recombinant Na⁺/K⁺-ATPase, we performed whole-cell patch-clamp[14] using HEK 293 F cells that over-express α3/β1/FXYD6. Indeed, the transduced HEK 293 F cells show ~4-fold enhanced pump activity, 2.17 ± 0.90 pA/pF as compared with background pump-activity in mock-transduced cells, 0.57 ± 0.06 pA/pF (Fig. 1c, Supplementary Fig. 1c). For this recombinant Na⁺/K⁺-ATPase, the number of phosphorylation sites measured is 0.95 ± 0.09 nmol/ mg protein (Fig. 1d) and the ATP turnover rate measured is 252 ± 14 µmol Pᵢ/mg protein/hour (Fig. 1e), which are comparable to those of the tissue-prepared endogenous Na⁺/K⁺-ATPases[15].

### Structural determination of the human Na⁺/K⁺-ATPase in its Na⁺- and K⁺- occluded states

We trapped the human α3 Na⁺/K⁺-ATPase in its Na⁺-occluded state using a buffer containing Na⁺ and ADP in conjunction of AlF₄⁻, and its

K$^+$-occluded state using K$^+$ and MgF$_4^{2-}$ (see Methods for more details). The single-particle cryo-EM analysis yielded 3.7-Å-resolution Na$^+$-occluded (Fig. 2a) and 4.1-Å-resolution K$^+$-occluded (Fig. 2b) maps. The cryo-EM densities of the ADP-AlF$_4^-$ (Fig. 2c) and MgF$_4^{2-}$ (Fig. 2d) in the cytoplasmic domains in both maps are well-defined. The cryo-EM densities of the two maps are sufficient to build reliable models for the structures of the human α3 Na$^+$/K$^+$-ATPase in the Na$^+$-occluded (E1·P-ADP) (Fig. 2e) and K$^+$-occluded (E2·Pi) (Fig. 2f) states, with the help of structures of the pig Na$^+$/K$^+$-ATPase (PDBs 3WGU and 3B8E).

Architecturally, the structure of the human sodium-potassium pump is virtually identical to that of other Na$^+$/K$^+$-ATPases, which were previously determined by X-ray crystallography[5,7,9,10,16]. α3, β1 and FXYD6 form a stable complex assembly through the interactions of their TMDs. In particular, the 45°-tilted single TM of the β1 and the single TM of the FXYD6 interact with the α3's TMD through M7/M10 helices and M9 helix, respectively (Fig. 2e, f). Of the α3's TMD, the M1–M6 are conserved among P-type ATPase superfamily members and form the minimal transport domain while the TMD's M7–M10 function as membrane anchor and support the transport domain[2]. The A domain directly connects to the TMD through M1–M3, while the N domain links to the P domain which directly connects to the TMD through M4–M5 (Fig. 2e, f). These connections couple the conformational rearrangements between the cytoplasmic domains in different phosphorylation/ dephosphorylation states and the transmembrane domain in different substrate-binding modes[2,17].

As anticipated, in the presence of Na$^+$ and ADP in conjunction of AlF$_4^-$, the human α3 Na$^+$/K$^+$-ATPase is trapped in the Na$^+$-occluded state (E1·P-ADP). Superposition of the Na$^+$-occluded structures of the human α3 (this study) and the pig α1 Na$^+$/K$^+$-ATPase (PDB: 3WGU) by aligning their alpha subunit showed no major structural difference with a root-mean-square deviation (r.m.s.d) ~ 0.8 Å (Supplementary Fig. 1d). The transmembrane helices M1–M6 adopt a typical occluded conformation as seen in the pig ortholog. Like the Na$^+$-occluded state, the K$^+$-occluded-state (E2·P$_i$) structure of the human α3 (this study) is nearly identical to that of the pig α1 ortholog (PDB: 3B8E) (r.m.s.d ~ 0.8 Å) (Supplementary Fig. 1e). Despite the strong densities of all the helices in the TMDs, except Q920, other highly conserved cation-binding residues, including E324 (M4), N773 and E776 (M5), D801 and D805 (M6) are poorly resolved in the occluded-state cryo-EM maps (Fig. 2g, h) due to radiation damage[18]. Nevertheless, we can model the binding cavity and bound cations reliably using the coordinates from the previously determined occluded structures of the pig Na$^+$/K$^+$-ATPase (Supplementary Fig. 1f, g).

## Stabilization and structural determination of the fully exoplasmic side-open and cytoplasmic side-open states

We speculated that the reason why Na$^+$/K$^+$-ATPase favorably adopts an Na$^+$-occluded or K$^+$-occluded ligand-bound conformations is due to the co-purified Na$^+$ or K$^+$ ions. To trap Na$^+$/K$^+$-ATPase in its non-occluded states, we substituted the four cation-binding residues (E324, E776, D801 and D805) of the α3 (Supplementary Fig. 1f, g) into alanine to completely diminish the affinities of Na$^+$ and K$^+$ binding. This cation-binding deficient Na$^+$/K$^+$-ATPase, termed "4A mutant" has relatively high surface expression (Supplementary Fig. 1h), but exhibits significantly reduced pump activity (0.88 ± 0.19 pA/pF) (Fig. 1c, Supplementary Fig. 1c). In consistency with the reduction in transport activity, the 4A mutant reduces both its phosphorylation activity to 0.29 ± 0.07 nmol sites/mg protein (Fig. 1d) and its ATP turnover rate to 17 ± 5 μmol P$_i$/mg protein/hour (Fig. 1e). These observations suggested that the 4A mutant has correct protein folding; however, its enzymatic activities, including Na$^+$-induced phosphorylation and K$^+$-induced ATP hydrolysis become less sensitive to Na$^+$ and K$^+$, respectively. Using this cation-binding deficient 4A mutant Na$^+$/K$^+$-ATPase, we determined three cryo-EM structures of the Na$^+$/K$^+$-ATPase in its exoplasmic side-open state at 3.9-Å resolution, as mimicked the E2P state, and its cytoplasmic side-

open states at 3.4-Å and 3.5-Å resolutions, as mimicked the E1 and E1·ATP, respectively.

## Exoplasmic-gating mechanism

The structure of the exoplasmic side-open human α3 (Fig. 3a, b) is nearly identical to that of the pig kidney α1 Na$^+$/K$^+$-ATPase (r.m.s.d. ~ 0.9 Å) stabilized in the E2P conformation by BeF$_3^-$ bound to the Aspartate 366 residues of the P domain (Supplementary Fig. 2a)[11,13]. All the subunits and ten transmembrane helices of the human α3 subunit overlap perfectly with those of the pig kidney α1 subunit (Supplementary Fig. 2b, c) confirming that the cation-binding deficient 4A mutant is functionally and structurally relevant.

To study the exoplasmic-gating mechanism of the Na$^+$/K$^+$-ATPase, we superimposed the structures of the exoplasmic side-open and K$^+$-occluded states by aligning the M7–M10 helices. This comparison revealed the M1–M4 helices undergo major conformational rearrangements between the K$^+$-occluded and the exoplasmic side-open state while minor changes were observed in the M5–M10 (Supplementary Fig. 2d–f). In the exoplasmic side-open state, the A domain tilts ~7° along the vertical axis in the presence of BeF$_3^-$ relative to that of the K$^+$-occluded state (Fig. 3c). Along with the conformational change of the A domain, the M1–M2 and M3–M4E helices rotate ~15 ° and 14°, respectively horizontally as a whole rigid body (Fig. 3c) in the K$^+$-occluded compared to those in the exoplasmic side-open states. The M2 is unwound in the K$^+$-occluded state, but forms a well-folded α-helix in the exoplasmic side-open state. Such conformational change of M2, in turn, drives its downward movement (Supplementary Fig. 2e). Most of the key K$^+$-binding residues, including N773 and E776 (M5), D801 and D805 (M6) undergo minor rearrangements (Supplementary Fig. 2g) between the two states.

The exoplasmic gate of the Na$^+$/K$^+$-ATPases is defined by the arrangement of the M4E relative to the M6 helix (Fig. 3d). In the K$^+$-occluded state, M4E and M6 helices are in a proximal distance. Particularly, V319 of M4E and L798 of M6 (7.7 Å Cα–Cα distance) are situated closely with each other (Fig. 3d) and localized on top of the potassium binding cavity, serving as the exoplasmic-gating latch closing the K$^+$ entry pathway (Fig. 3e). In the exoplasmic side-open state, the rotation of M4E distances V319 from L798 of M6 helix (12.6 Å Cα–Cα distance) (Fig. 3d) and thereby creates an open K$^+$ entry pathway (Fig. 3f) along M4E and M6 helices. To study the role of the exoplasmic-gating latch residue V319 (M4E), we mutated it to alanine. This mutation has minor effect on the phosphorylation activity, 0.84 ± 0.15 nmol sites/ mg protein compared to the WT, 0.95 ± 0.09 nmol sites/ mg protein (Fig. 3g). Interestingly, V319A mutation significantly increases the maximal ATPase activity of the Na$^+$/K$^+$-ATPase (Figs. 3h, 4i). However, while the V319A mutant remains its apparent affinity toward K$^+$, 0.5 ± 0.1 mM, compared to that of the WT, 0.4 ± 0.1 mM (Fig. 3h), its apparent affinity toward Na$^+$ significantly reduces, 24.0 ± 3.3 mM compared to that of the WT, 9.2 ± 1.3 mM (Fig. 4i). These data indicated that V319A mutation likely shifts its E1-E2 equilibrium toward E2 state that prefers K$^+$ ions to Na$^+$ ion binding, thus increases the dephosphorylation rate of the Na$^+$/K$^+$-ATPase.

## Cytoplasmic-gating mechanism

The well-defined 3.4 Å and 3.5 Å maps of the human α3 4A mutant stabilized in its cytoplasmic side-open state in the absence or presence of AMPPMP enabled us to build reliably the atomic models of the human α3 Na$^+$/K$^+$-ATPase in its cytoplasmic-side open (E1) (Fig. 4a, b) and AMPPMP-bound cytoplasmic-side open (E1·ATP) (Fig. 4c, d). Superposition of the Na$^+$/K$^+$-ATPase in the cytoplasmic side-open and Na$^+$-occluded states by the TM7-10 helices revealed that major conformational changes occur at the α3 subunit while the β1's and FXYD6's conformations remain unchanged (Supplementary Fig. 3a). Within the TMD of the α3, M1–M4 helices undergo major rearrangements while minor conformational changes occur at M5–M10 helices

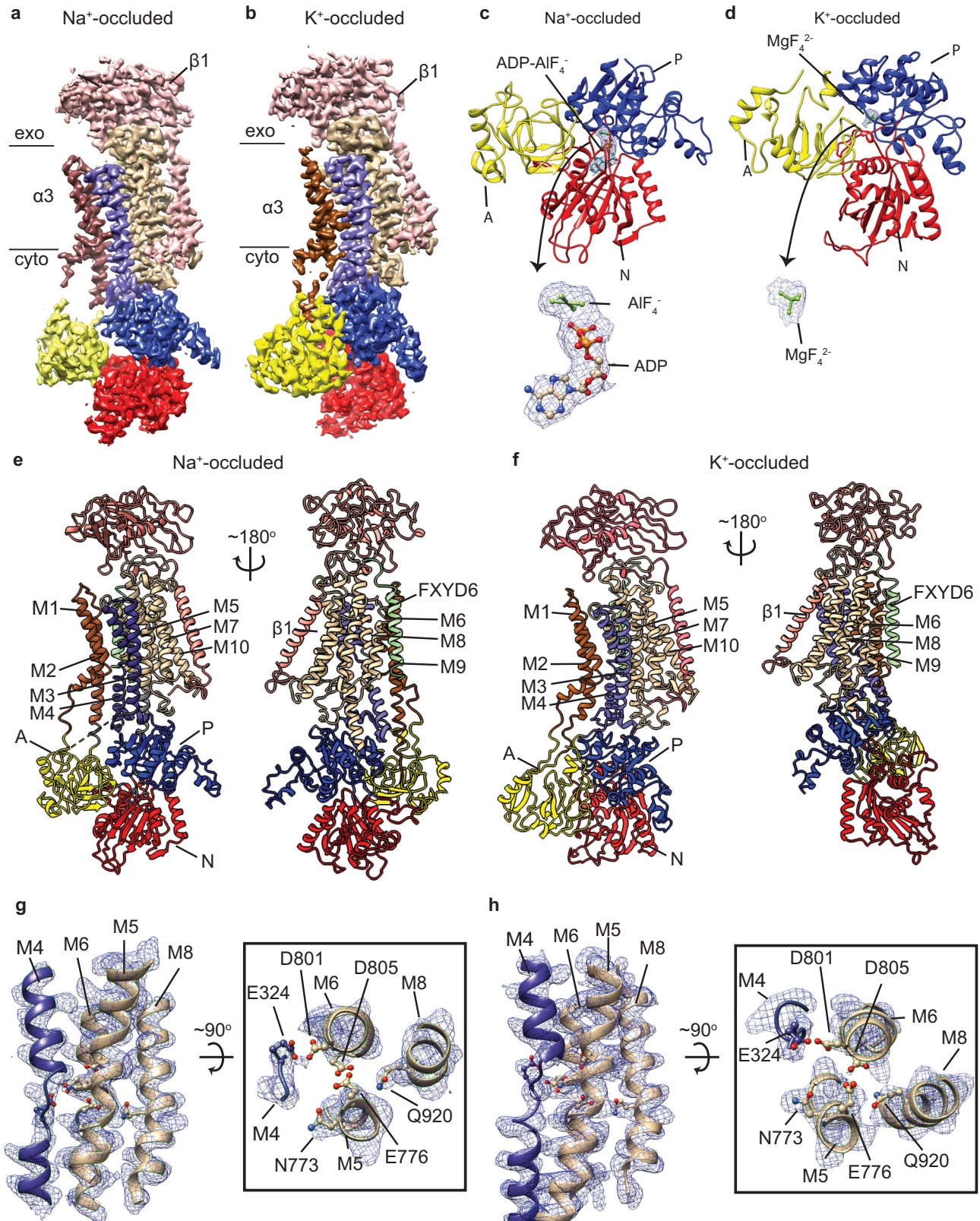

**Fig. 2 | Cryo-EM maps and structures of the human α3 Na⁺/K⁺ ATPase in its Na⁺- and K⁺-occluded states. a** 3.7 Å- resolution cryo-EM map of the human α3 Na⁺-occluded structure. exo: exoplasm, cyto: cytoplasm. **b** 4.1 Å- resolution cryo-EM map of the human α3 K⁺-occluded structure. exo: exoplasm, cyto: cytoplasm. **c** ADP-AlF₄⁻ trapped cytoplasmic domains of the Na⁺-occluded structure. The mesh represents electron density of ADP-AlF₄⁻ **d** MgF₄²⁻ trapped cytoplasmic domains of

the K⁺-occluded-state structure. The mesh represents electron density of MgF₄²⁻. **e** Structural model of the human α3 Na⁺/K⁺ ATPase in its Na⁺-occluded state. The black dashed line represents missing density of amino acid residues 261–270 in the structural model. **f** Structural model of the human α3 Na⁺/K⁺-ATPase in its K⁺-occluded state. **g**, **h** Densities of the cation-binding residues and their respective helices in the Na⁺-occluded (**g**) and K⁺-occluded structure (**h**).

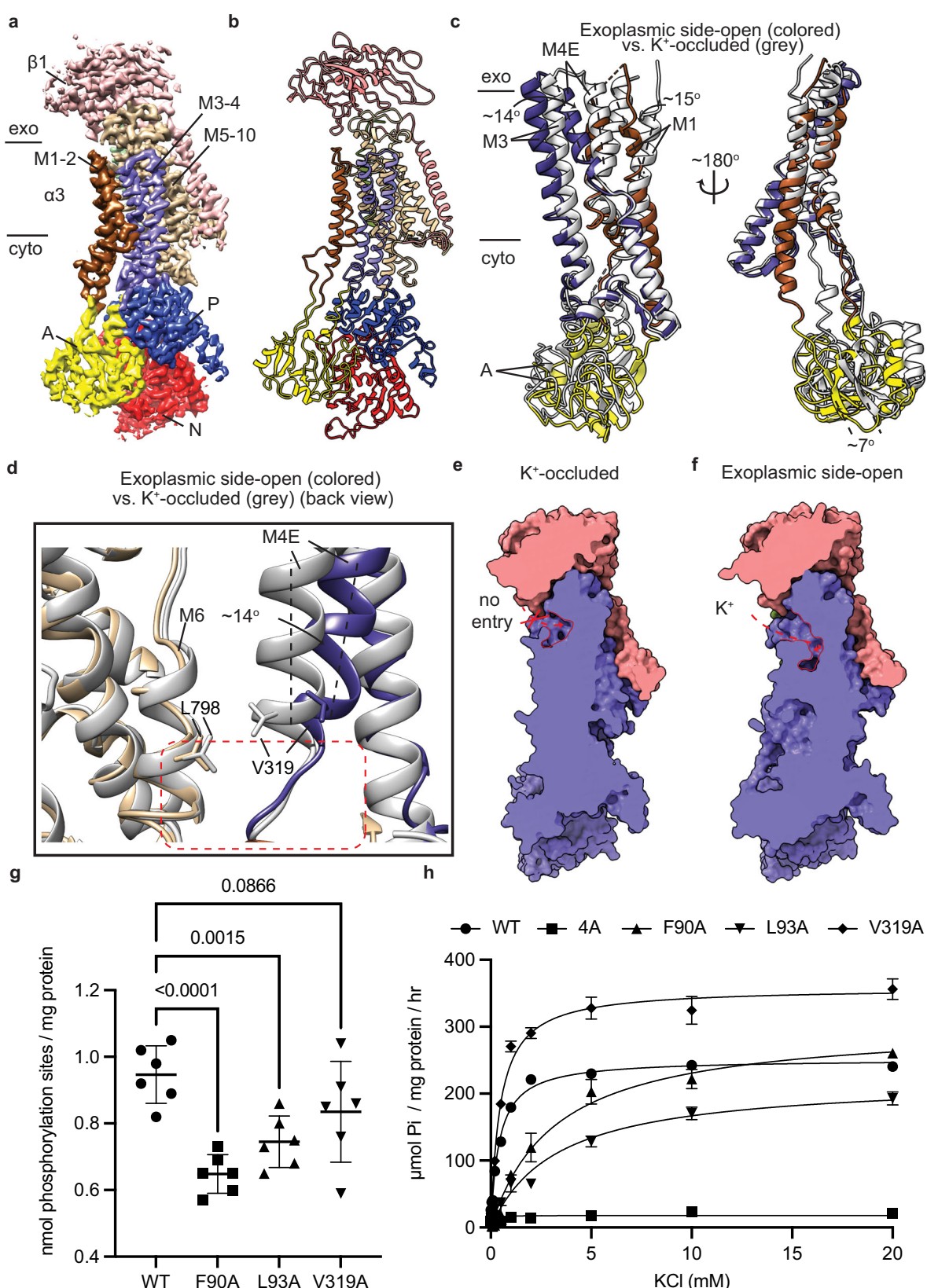

(Supplementary Fig. 3a–c). Except E324 in M4, most of the Na⁺ binding residues situated in the unmoving part of the TMD, including helices M5 (N773 and E776), M6 (D801 and D805), and M8 (Q920). Hence, these cation-binding residues only move slightly within the Na⁺ binding pockets between the cytoplasmic side-open and the Na⁺-occluded state to accommodate for Na⁺ binding (Supplementary Fig. 3d).

The rearrangement of M1–M4 helices couples with the conformational change of the cytoplasmic A domain. Specifically, the A domain rotates ~14° horizontally along the axis perpendicular to the plasma membranes in the cytoplasmic side-open state relative to that in the Na⁺-occluded state (Fig. 4e). As the cytoplasmic A domain is connected to M1–M3 through three rigid linkers, this rotation in the A

**Fig. 3 | 3.9 Å-resolution structure of the human α3 Na⁺/K⁺ ATPase in its exo-plasmic side-open state. a, b** Cryo-EM map (**a**) and structural model (**b**) of the human α3 Na⁺/K⁺-ATPase in its exoplasmic side-open state (E2P). exo: exoplasm, cyto: cytoplasm. **c** Superimpose the M1–M2 helices (brown), M3–M4 helices (slate) and A domain (yellow) of the human α3 exoplasmic side-open (colored) and K⁺-occluded (gray) structures by aligning their M7–M10 helices. **d** Superimpose K⁺ entry pathway in the exoplasmic side-open (colored) and K⁺-occluded (gray) structures of the human α3. Back view: FXYD6 was shown in the front of the structural model. **e** Slab view of the K⁺-occluded structure shows no K⁺ entry

pathway (red trace). **f** Slab view of the human exoplasmic side-open structure shows the K⁺ entry pathway (red trace). **g, h** the numbers of phosphorylation sites (**g**) and K⁺-dependent ATPase activities (**h**) of the wild-type (WT) Na⁺/K⁺-ATPase and its mutants, including F90A, L93A and V319A. $N = 6$ independent experiments. Data were presented as mean ± standard deviation (SD). One-way ANOVA test was used to compare mean values of multiple different variants. $P$-values were shown as numerical numbers in the subfigure 3 g. Source data are provided as a Source Data file.

domain induces a ~19° outward movement of M1 between the cyto-plasmic side-open and Na⁺-occluded states. Our observation is con-sistent with early proteolytic digestion experiments by Jorgensen showing that the movement of the lysine-rich N-terminus that directly connects to M1 helix of the alpha subunit plays a crucial role in the transition from the E2 to the E1 state of the Na⁺/K⁺-ATPase[19].

In the Na⁺-occluded state, F90 and L93 are localized in close proximity to T286 (M3) (10.2 and 9.9 Å Cα–Cα distances, respectively) (Fig. 4f), closing the Na⁺ entry pathway from the cytoplasm (Fig. 4g). While F90 residues acts at the first gating latch that regulates Na⁺ access into the transmembrane region between M1 and M3 helices, L93 situates right next to the Na⁺ binding sites, serving as a second gating latch and supporting the M3 helix in the Na⁺-occluded state (Fig. 4f). In contrast, F90 and L93, in the cytoplasmic side-open state, are further away from T286 of M3 (14.2 and 17.8 Å Cα–Cα distance, respectively) (Fig. 4f), forming a wide-open cavity at the cytoplasmic gate in the cytoplasmic side-open state (Fig. 4f). To study the roles of these cytoplasmic gating residues, we mutated the two residues F90 and L93 to alanine. Both F90A and L93A mutations slow the rates of Na⁺-dependent phosphorylation (Fig. 3g). Additionally, the two muta-tions lower their apparent affinities toward K⁺, 2.4 ± 0.5 mM (F90A) and 3.3 ± 0.4 mM (L93A) compared to that of the WT, 0.4 ± 0.1 mM (Fig. 3h), leading to decreased dephosphorylation rates. Hence, at low K⁺ concentrations, F90A and L93A mutants exhibit significantly lower ATP turnover rates, as compared to that of the WT (Fig. 3h). These results together suggest that both F90 and L93 residues are critical for the functionality of Na⁺, K⁺-ATPase, which strongly supports our structural model.

In addition, the M1 and M2 helices also undergo a one-helix-turn screw-like movement along their helix axis toward the membranes while the M3 vertically shifts upward by ~3 Å (Supplementary Fig. 3b). The M4 helix is directly connected to the P domain, so its movement is dictated by the P domain's rearrangement. The cytoplasmic side-open M4E (M4's residues 309–320) is slightly tilted compared to that in the Na⁺-occluded state (Supplementary Fig. 3b). The movements descri-bed above potentially expand the Na⁺ entry pathway and facilitate Na⁺ access into the binding pockets of the cytoplasmic side-open Na⁺/K⁺-ATPase.

In the E1•ATP structure, the cytoplasmic domains are stabilized by the AMPPCP bound to the interface of N and P domains of the α3 subunit (Supplementary Fig. 3e). Superposition of the cytoplasmic domains in the E1•ATP and E1•P-ADP states by the P domain showed that a helix formed by residues 708-717 of the P domain rotates about ~14° closer to the N domain in the E1•P-ADP compared to that in E1•ATP state (Supplementary Fig. 3f). This rotation creates a hydrogen bond between residue N710 and the gamma phosphate group of the ATP molecule which may facilitate the autophosphorylation reaction of the human α3.

## Discussion

In this work, we focused on structural and functional characterization of the human α3 Na⁺/K⁺-ATPase that comprises α3, β1 and FXYD6. We reported the structures of the human α3 Na⁺/K⁺-ATPase in five distinct intermediate states, of which the cytoplasmic side-open-state struc-tures fill a gap in the structural understanding of the Na⁺/K⁺-ATPase's

transport cycle and gating mechanism of the Na⁺/K⁺-ATPase. To cap-ture the cytoplasmic side-open states, we introduced a quadruple alanine mutation variant (4A mutant) at the key ion binding residues to diminish the ion-binding capacity of the Na⁺/K⁺-ATPase. We anticipated that our approach may lead to unforeseen changes in the structure of the Na⁺/K⁺-ATPase. Therefore, we performed functional and structural characterization on the 4A mutant. Our functional data suggests that the 4A mutant still retained a certain level of its phosphorylation and ATPase activities (Fig. 1d, e). In consistency with the functional data, our structural data also showed a virtually identical structure of the human α3 4A mutant and the pig α1 WT ATPase in the exoplasmic side-open state (Supplementary Fig. 2b). These evidences confirmed that the 4A mutant was appropriate to use for solving the cytoplasmic side-open states of the Na⁺/K⁺-ATPase.

Superposition of the M1–M6 helices between the cyto-plasmic side-open and the exoplasmic-side open states revealed the gating mechanism of the Na⁺/K⁺-ATPase exists in each state of the Na⁺/K⁺-ATPase during its transport cycle; for instance, in the cytoplasmic side-open state, the cytoplasmic gate is open, while the exoplasmic gate is closed, whereas in the exoplasmic side-open state, the exoplasmic gate is open while the cytoplasmic gate is closed (Supplementary Fig. 4).

### Comparison of gating mechanisms of P2 ATPases

Our structural insights of the human α3 Na⁺/K⁺-ATPase in its exoplasmic-side open state re-emphasizes the rearrangement of the M4E[20] relative to the M6 helix to open the ion pathway toward the exoplasm. Interestingly, the M1–M4 rearrangement in the exo-plasmic side-open state is similar to that of the Na⁺/K⁺-ATPase in dif-ferent cardiac glycoside-bound structures[6,8,11,21] (Supplementary Fig. 5a) and the gastric proton pump type-2 H⁺/K⁺-ATPases in its blocker-bound conformation[22]. A similar rotation of the M3–M4 was also observed in sarcoendoplasmic reticulum Ca²⁺-ATPase (SERCA) in its luminal side-open state compared to its Ca²⁺-occluded state[23,24] (Supplementary Fig. 5b). These similar conformational changes sug-gest that the exoplasmic-gating mechanism is conserved among P2 ATPase family.

Prior to our work, the SERCA was the only member of the P2 ATPase family whose cytoplasmic gating mechanism was functionally and structurally well-characterized[23,24]. Our cytoplasmic side-open structures of the human α3 Na⁺/K⁺-ATPase reveal a different operation of the cytoplasmic gate compared to that of the SERCA. In particular, to expose the Ca²⁺ binding sites in the transmembrane domain, the SERCA's M1 (L61) moves vertically along its own axis by 12 Å (two-helix turns) into the luminal membranes as a "sliding-door" mechanism (Supplementary Fig. 5c). On the other hand, the Na⁺/K⁺-ATPase uses a "hinged-door" 19 Å outward rotation of M1 (F90 and L93) distal from M3 (T286) to create a wide-open cytoplasmic gate for Na⁺ access in the human α3 Na⁺/K⁺-ATPase (Fig. 4f). We speculate these different operations of the two M1 helices are due to the different movements of the A domains that directly link to the M1 helices of the two ATPases. In particular, while the SERCA's A domain moves up ~12 Å along a vertical axis (perpendicular to the lipid bilayers) (Supplementary Fig. 5c) pushing the M1 toward the lumen membrane, the Na⁺/K⁺-ATPase's A domain rotates outward relative to the P domain in the E1 and E1•ATP

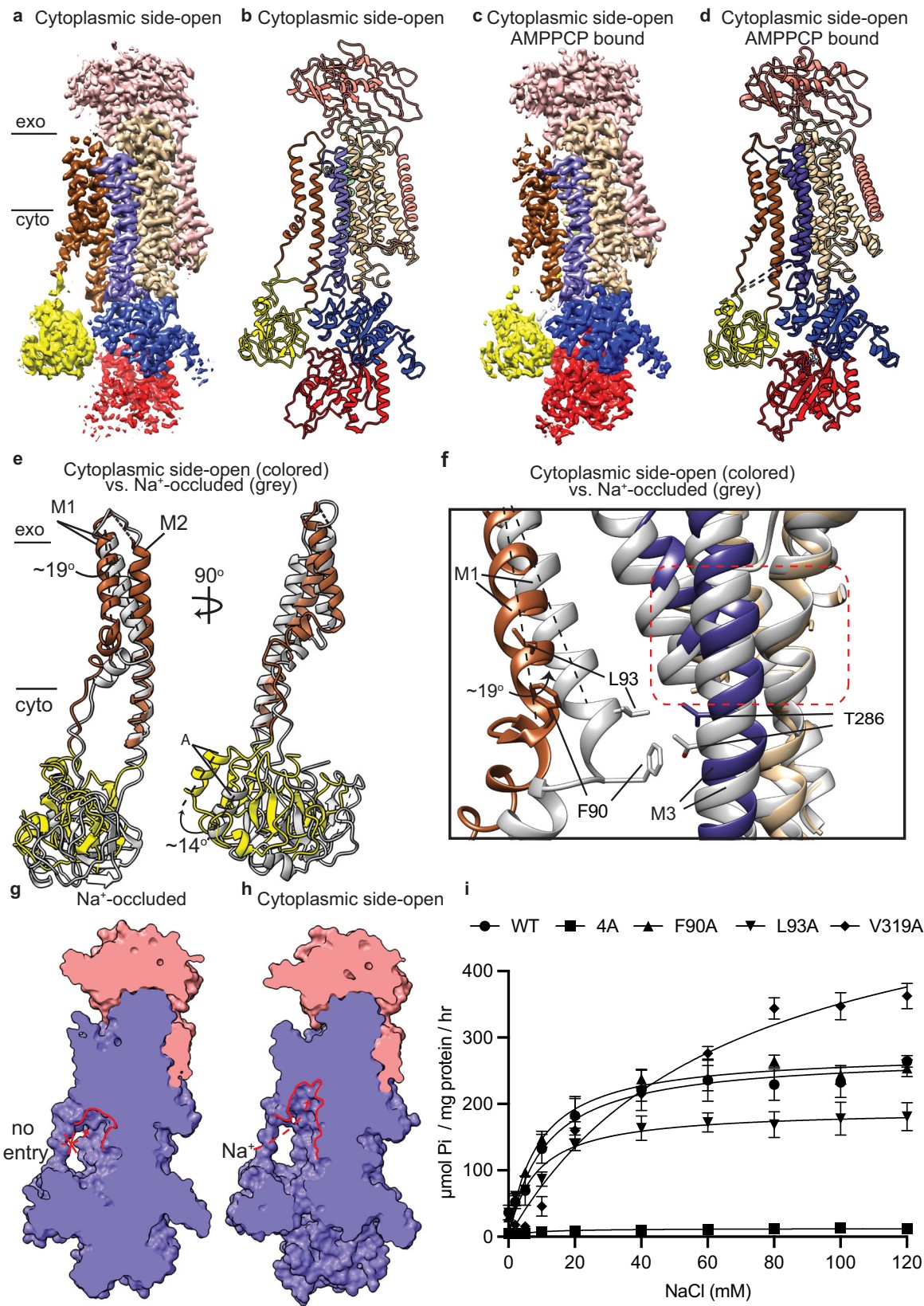

structures. In addition, while the SERCA comprises a single alpha subunit, the Na⁺/K⁺-ATPases compose of multiple subunits, including alpha, beta and FXYD. We can't rule out the possibility that some structurally unresolved N- and C-terminal regions of the beta and FXYD subunits may play a certain roles in regulating the cytoplasmic gate of the Na⁺/K⁺-ATPases. Hence, further functional studies need to be done

to understand any potential role(s) of the beta and FXYD subunits in the cytoplasmic gating mechanism of the Na⁺/K⁺-ATPases.

### A structural model for gating mechanism of the Na⁺/K⁺-ATPase

Our work demonstrates a complete transport cycle for transporting Na⁺/K⁺ with different intermediate states of the human α3

**Fig. 4 | 3.4 Å and 3.5 Å- resolution structures of the human α3 Na⁺/K⁺-ATPase in its cytoplasmic side-open state. a, b** Cryo-EM map (**a**) and structural model (**b**) of the human α3 Na⁺/K⁺-ATPase in its cytoplasmic side-open state (E1). exo: exoplasm, cyto: cytoplasm. **c, d** Cryo-EM map (**c**) and structural model (**d**) of the Na⁺/K⁺-ATPase in its AMPPCP-bound cytoplasmic side-open state (E1•ATP). The black dashed line represents missing density of amino acid residues 260–271 in the structural model. (**e**) Superimpose the M1–M2 helices and A domain of the cytoplasmic side-open (colored) and Na⁺-occluded (gray) structures of the human α3 by aligning their M7–10 helices. **f** Superimpose Na⁺ entry pathway in the cytoplasmic

side-open (colored) and Na⁺-occluded (gray) structures of the Na⁺/K⁺-ATPase. Front view: FXYD6 was shown in the back of the structural model. **g** Slab view of the Na⁺-occluded Na⁺/K⁺-ATPase shows no Na⁺ entry pathway (red trace). **h** Slab view of the cytoplasmic side-open Na⁺/K⁺-ATPase shows the Na⁺ entry pathway (red trace). **i** Na⁺-dependent ATPase activity of the wild-type (WT) Na⁺/K⁺-ATPase compared to its mutants, F90A, L93A, V319A and 4A mutants. $N = 6$ independent experiments. Data were presented as mean ± standard deviation (SD). One-way ANOVA test was used to compare mean values of multiple different variants. Source data are provided as a Source Data file.

Na⁺/K⁺-ATPase (Supplementary Fig. 6). With an emphasis on revealing the structural insights for gating mechanism of the Na⁺/K⁺-ATPase, we generated a model focusing on the M1–M6 helices, with M7–M10 helices removed for better illustration (Fig. 5). The gating regulation of the Na⁺/K⁺-ATPase is done at two distinct cytoplasmic and exoplasmic gates controlling directionality of ion trafficking into and out of Na⁺/K⁺-ATPase at specific intermediate states. The cytoplasmic gate is regulated by M1 and M3, while the exoplasmic gate is regulated by M4E and M6. The transport begins with an open cytoplasmic gate created by the distal arrangement of M1 and M3 allowing Na⁺ ions to enter the cytoplasmic gate, while the M4E and M6 form a closed exoplasmic gate preventing K⁺ or non-specific ions to enter from the exoplasm. Both gates are closed to secure Na⁺ in the ion-binding pockets in the Na⁺-occluded state where autophosphorylation occurs. ADP release triggers a large rotation of the cytoplasmic A domain, inserting its TGES motif into the nucleotide binding cavity, previously formed by the N and P domains in the E1•P-ADP state (Supplementary Fig. 7a, b). This rotation induces conformational changes in the M1 and M2, and subsequently M4E and M6 (Supplementary Fig. 7c) to open the exoplasmic gate, releasing Na⁺ ions and allowing K⁺ binding in the exoplasmic side-open state. In this state, the M1 and M3 remain proximal, closing the cytoplasmic gate to prevent Na⁺ or other ions to pass through, while the K⁺ ions are selected at the K⁺ binding sites. Upon K⁺ binding, M4E and M6 close their K⁺ entry pathway and K⁺ ions are secured inside the K⁺- binding pockets. Dephosphorylation triggers the opening of the cytoplasmic gate to release K⁺ into the cytoplasm and the cycle restarts (Fig. 5).

In summary, we report five structures representing different intermediate states of the human α3 Na⁺/K⁺-ATPase during its transport cycle. Our finding suggests the structural insights for gating mechanism of the Na⁺/K⁺-ATPase (Fig. 5) and describes a complete transport cycle of the Na⁺/K⁺-ATPase (Supplementary Fig. 6). Our work provides opportunities to understand both the structure-function relationship and pathophysiology of the human α3 Na⁺/K⁺-ATPase.

## Methods
### Constructs and cell growth
Full-length human N-terminal Flag-tagged *ATPA3*, *ATP1B1*, and *FXYD6* cDNAs were sub-cloned into pEZT-BM vector[25] (Addgene) with ribosome skipping sequences (P2A) (pEZT-NKA construct). Mutations, including substrate-binding-deficient 4 A (E327A, E776A, D801A, and D805A), cytoplasmic-gating-deficient F90A and L93A, and exoplasmic-gating-deficient V319A were introduced by site-directed mutagenesis. The Na⁺/K⁺-ATPase was expressed in human embryoid kidney (HEK) 293 F cells (Invitrogen, cat# R79007) using the bac-mam expression system as described elsewhere[26–29]. Briefly, a single bacmid carrying the three genes encoding for α3, β1, FXYD6 were generated by transforming the pEZT-NKA plasmid to the *E. coli* strain DH10Bac. Baculoviruses, produced by transfecting *Spodoptera frugiperda* (Sf9) cells (Thermo Fisher Scientific, cat# 11496015) with the bacmid, were used to infect FreeStyle TM HEK 293-F cells (~3 × 10⁶ cells/ ml). Cells were cultured at 37 °C, 5% CO₂ for 12 h before added with 10 mM sodium butyrate to boost protein expression[30] and switched to 30 °C, 5% CO₂ for 60 h. Cells were then harvested by centrifugation at 4000 × g at

4 °C for 20 min. Cell pellets were homogenized in hypotonic buffer containing 10 mM HEPES-Na pH 7.4, 10 mM NaCl, 10% glycerol, 0.2 mM PMSF and protease inhibitors (1 μg/ml leupeptin, 1 μg/ml pepstatin, 1 mM benzamidine HCl, 1 μg/ml aprotinin, 100 μg/ml trypsin inhibitor from soybean (Sigma)), flash-frozen by liquid nitrogen and stored at −80 °C.

### Protein purification
Homogenized cells were thawed and centrifuged at 4000 × g at 4 °C for 20 min. Supernatants containing cytosolic lysate were discarded. Pellets were resuspended into lysis buffer containing 20 mM HEPES-Na pH 7.4, 250 mM NaCl, 10% (v/v) glycerol, 1% (w/v) LMNG and incubated at 4 °C for 1 h. The unlysed cells, debris and excess membranes were removed by centrifuging at 48,000 × g for 30 min at 4 °C. The Na⁺/K⁺-ATPase protein was captured by affinity purification using Flag M2 resins (Sigma), eluted by buffer containing 200 μg/ ml Flag peptide (Fisher Scientific) and further purified by size exclusion chromatography equilibrated in a buffer containing 20 mM HEPES-Na pH 7.4, 150 mM NaCl, 1% LMNG (w/v) or 20 mM HEPES-K pH 7.4, 150 mM KCl, 1% (w/v) LMNG. The LMNG-solubilized Na⁺/K⁺-ATPase was used for ATPase activity assays. Proteins used for cryo-EM sample preparation were purified in the same manner with a minor modification. Detergent was exchanged from LMNG to digitonin during the affinity purification and size exclusion chromatography using a buffer containing 20 mM HEPES-Na pH 7.4, 150 mM NaCl, 0.06% (w/v) Digitonin; or 20 mM HEPES-K pH 7.4, 150 mM KCl, 0.06% (w/v) Digitonin.

### Sample preparation and EM data acquisition
To trap the human α3 Na⁺/K⁺-ATPase in its Na⁺-occluded state (E1•P-ADP), the WT Na⁺/K⁺-ATPase was incubated in a buffer containing 20 mM HEPES-Na pH 7.4, 150 mM NaCl, 1 mM MgCl₂, 8 mM NaF, 2 mM AlCl₃ and 0.06% (w/v) digitonin at room temperature for 1 h. Similarly, a buffer containing 20 mM HEPES-K pH 7.4, 150 mM KCl, 1 mM MgCl₂, 8 mM NaF and 0.06% (w/v) digitonin was used to trap the protein in its K⁺-occluded state (E2•Pi).

The 4A mutant α3 Na⁺/K⁺-ATPase in its purification buffer containing 20 mM HEPES-Na pH 7.4, 150 mM NaCl and 0.06% (w/v) digitonin was used to capture its cytoplasmic side-open state (E1). A supplement of 1 mM AMPPCP pH 7.4 and 4 mM MgCl₂ was used to trap the protein in its ATP-bound cytoplasmic side-open state (E1•ATP). The mixture was incubated on ice for 1 h.

To trap the exoplasmic side-open state (E2P), a supplement of 1 mM BeSO₄, 4 mM NaF and 1 mM MgCl₂ was added to the 4A mutant Na⁺/K⁺-ATPase. The mixture was incubated at room temperature for 3 h.

All the proteins samples above were spun down at 18,000 × g at 4 °C for 30 min before subjected to the cryo-EM grids. The cryo-EM grids were prepared by applying 3 μl of protein samples (4 mg/ml) to glow discharged Quantifoil R1.2/1.3 300-mesh gold holey carbon grids (Quantifoil, Micro Tools GmbH, Germany). Grids were blotted for 4.0 s under 100% humidity at 4 °C before being plunged into liquid ethane using Mark IV Vitrobot (FEI).

Micrographs were acquired on a Titan Krios microscope (FEI) operated at 300 kV with a K3 Summit direct electron detector

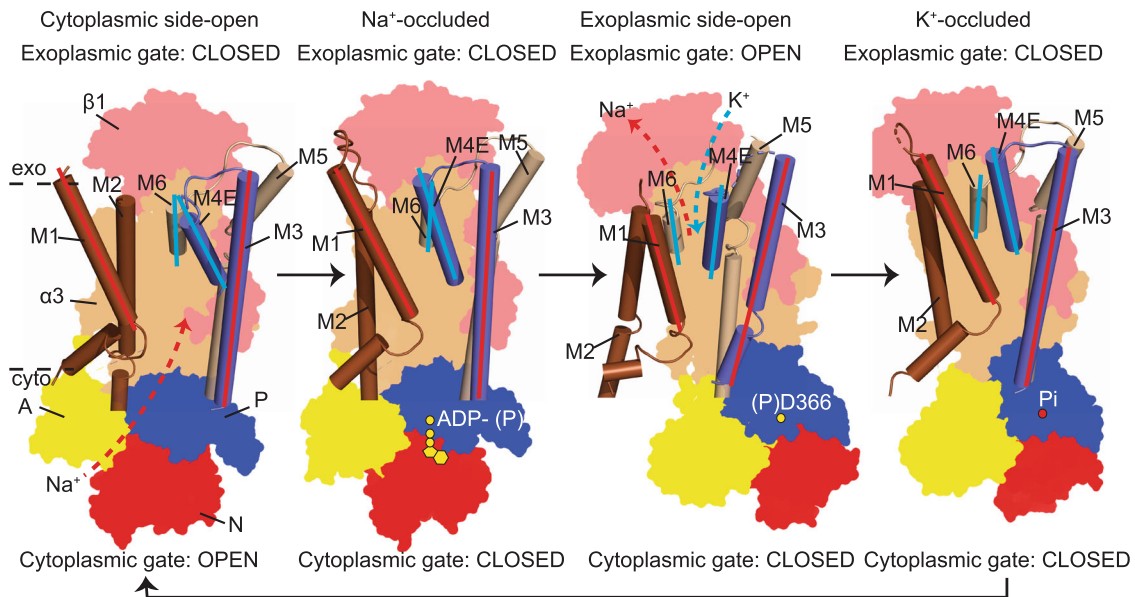

**Fig. 5 | A proposed gating mechanism of the Na⁺/K⁺ ATPase.** A schematic diagram demonstrates the gating mechanism of the Na⁺/K⁺-ATPase at its cytoplasmic and exoplasmic sides. Both cytoplasmic gate, regulated by M1 and M3 helices, and exoplasmic gate, regulated by M4E and M6 work in concert to permit Na⁺/K⁺ access at a certain state of the Na⁺/K⁺-ATPase. Specifically, in the cytoplasmic side-open state, the cytoplasmic gate opens to allow Na⁺ access, while the exoplasmic gate remains closed. Upon Na⁺ and ATP binding, the Na⁺/K⁺-ATPase adopts its Na⁺-occluded state whose both gates are closed to secure Na⁺ in the cation-binding pockets, triggering autophosphorylation. The release of ADP triggers the exoplasmic side-open state of the Na⁺/K⁺-ATPase whose exoplasmic gate opens, releasing Na⁺ out and accepting K⁺ in, while the cytoplasmic gate remains closed to prevent Na⁺ backflow. K⁺ binding induces the K⁺-occluded state whose both gates are closed to secure bound K⁺ cations. Dephosphorylation releases K⁺ into cytoplasm and resets the cycle.

(Gatan), using a slit width of 20 eV on a GIF-Quantum energy filter. SerialEM was used for automated data collection following standard FEI procedure[31]. A calibrated magnification of 46,296 or 60,241 was used for, yielding a pixel size of 0.83 Å or 1.08 Å on images, respectively. The defocus range was set from −1.6 μm to −2.6 μm. Each micrograph was dose-fractionated to 30 frames with a total dose of about 60 e⁻/Å².

**Image processing**

The cryo-EM datasets of all 5 different Na⁺/K⁺-ATPase samples were processed in identical manners. The detailed image processing workflows were shown in Supplementary Figs. 8–12. The cryo-EM statistics for five different datasets were summarized in Table 1. Briefly, movie frames were motion-corrected and binned two-fold, and dose-weighted using MotionCor2[32]. The CTF parameters were estimated using Gctf[33]. RELION-3[34] was used for the following processing. Particles were first roughly picked by using the Laplacian-of-Gaussian blob method, and then subjected to 2D classification. Class averages representing projections of Na⁺/K⁺-ATPase in different orientations were used as templates for reference-based particle picking. Extracted particles were binned three times and subjected to 2D classification. Particles from the classes with fine structural feature were selected for 3D classification using an initial model generated from a subset of the particles in RELION. Particles from one of the resulting 3D classes showing good secondary structural features were selected and re-extracted into the original pixel size. Subsequently, we performed finer 3D classification by using local search in combination with small angular sampling (3.7°), resulting one new good class with improved density for the entire complex. The finial 3D refinement with the selected particles was performed with a soft mask around the whole Na⁺/K⁺-ATPase complex. The resolution for the resulting cryo-EM map was further improved by CTF refinement and particle polishing. Local resolution was calculated in RELION3. Resolution was estimated by applying a soft mask around the protein density with the Fourier Shell Correlation (FSC) 0.143 criterion.

**Model building, refinement and validation**

Overall, of the 5 cryo-EM maps, the α3's TMD's helices, especially M1−M4 had clearly well-defined densities (Supplementary Fig. 13) that enabled de novo building, while α3's cytoplasmic domains, β1 and FXYD6 were built based on their homologous models[5,10]. In brief, the 3.7-Å-resolution Na⁺-occluded (E1•P-ADP) and 4.1-Å-resolution K⁺-occluded (E2•Pi) human α3 Na⁺/K⁺-ATPase structures were built based on the homologous structures of the pig Na⁺/K⁺-ATPase (PDBs 3WGU and 3B8E). The models were manually adjusted in Coot[35]. The 3.5-Å resolution cryo-EM map of E1•ATP state is resolved in the highest quality with many well-defined side chains density for all of the α3, β1 and FXYD6, which enabled de novo building of an accurate model for the human α3 Na⁺/K⁺-ATPase, with the help of the Na⁺-occluded structure described above. The TMD was well resolved in the 3.4-Å resolution cryo-EM map of E1 state; however, it showed poor densities for the P and N domains presumably due to the high flexibility of these domains in the absence of ATP molecule. We rigid-body fitted the P and N domains of the E1•ATP into the E1 map guided by a few well-resolved helices. The cryo-EM structure of the BeF₃⁻-bound exoplasmic side-open (E2P) state was built in a similar manner as that used for cytoplasmic side-open structures. Briefly, the TMD of α3 was de novo built, whereas other domains were modeled mainly by rigid-body fitting. Phenix's real space refinements were used to refine coordinates of all the models. MolProbity in Phenix was used to validate the models[36]. The FSCs between models and maps were computed using the "Comprehensive validation" function in Phenix[36].

**Phosphorylation activity**

Phosphorylation activity was determined in the absence of K⁺ as described elsewhere[15]. In brief, LMNG-solubilized Na⁺/K⁺-ATPase wild-type (wt) and mutant proteins were reacted with excessive amount of [γ-³²P]ATP in a reaction buffer containing 20 mM HEPES-Na pH 7.4, 150 mM NaCl, 0.125 mM EDTA, 0.02% (w/v) LMNG and 10 mM MgCl₂. The reaction was incubated at 0 °C for 20 s, then terminated by adding a stop solution containing 10% w/v trichloroacetic acid and 2 mM

**Table 1 | Cryo-EM data collection and model statistics**

| | Na+-occluded<br>8D3U<br>EMD-27164 | K+-occluded<br>8D3X<br>EMD-27167 | Exoplasmic side-open<br>8D3Y<br>EMD-27168 | Cytoplasmic side-open<br>8D3V<br>EMD-27165 | AMPPCP-bound cytoplasmic side-open<br>8D3W<br>EMD-27166 |
|---|---|---|---|---|---|
| PDB<br>EMDB | | | | | |
| **Data collection and processing** | | | | | |
| Voltage (kV) | 300 | 300 | 300 | 300 | 300 |
| Electron exposure (e$^-$/Å$^2$) | 60 | 60 | 60 | 60 | 60 |
| Defocus range (μm) | 1.6 –2.6 | 1.6–2.6 | 1.6–2.6 | 1.6–2.6 | 1.6–2.6 |
| Pixel size (Å) | 0.83 | 0.83 | 1.08 | 1.08 | 1.08 |
| Symmetry imposed | C1 | C1 | C1 | C1 | C1 |
| Initial particle images (no.) | 1,889,299 | 2,017,712 | 1,559,941 | 1,970,027 | 2,926,310 |
| Final particle images (no.) | 108,246 | 91,096 | 84,639 | 227,349 | 98,088 |
| Map resolution (Å) | 3.7 | 4.1 | 3.9 | 3.4 | 3.5 |
| FSC threshold | 0.143 | 0.143 | 0.143 | 0.143 | 0.143 |
| Map resolution range (Å) | 3.7–5.2 | 3.9–5.3 | 3.7–5.6 | 3.2–4.6 | 3.3–4.6 |
| **Refinement** | | | | | |
| Model composition | 10135 | 10217 | 9896 | 10019 | 10046 |
| Non-hydrogen atoms | 1289 | 1305 | 1298 | 1280 | 1280 |
| Protein residues | 2 (ALF, ADP) | 1 (MF4) | 0 | 0 | 1 (ACP) |
| Ligands | | | | | |
| Bonds (RMSD) | 0.003 (0) | 0.003 (0) | 0.003 (0) | 0.003 (0) | 0.003 (0) |
| Lengths (Å) (# > 4σ) | 0.652 (6) | 0.587 (3) | 0.570 (0) | 0.495 (0) | 0.571 (5) |
| Angles (°) (# > 4σ) | | | | | |
| Validation | 1.93 | 1.90 | 1.96 | 1.68 | 1.86 |
| MolProbity score | 11.11 | 12.13 | 10.11 | 6.63 | 9.56 |
| Clashscore | 0.00 | 0.00 | 0.00 | 0.00 | 0.00 |
| Rotamer outliers (%) | 0.00 | 0.00 | 0.00 | 0.00 | 0.00 |
| Cß outliers (%) | | | | | |
| Ramachandran plot | 0.00 | 0.00 | 0.00 | 0.00 | 0.00 |
| Outliers (%) | 5.39 | 4.39 | 6.75 | 4.57 | 5.20 |
| Allowed (%) | 94.61 | 95.61 | 93.25 | 95.43 | 94.80 |
| Favored (%) | | | | | |
| B factors (Å) | 76.90 | 64.28 | 73.26 | 45.39 | 40.85 |
| Protein (mean) | 57.14 | 37.07 | – | – | 33.72 |
| Ligand (mean) | | | | | |
| Model vs. Data | 0.81 | 0.78 | 0.81 | 0.79 | 0.80 |
| CC (mask) | 0.64 | 0.57 | 0.60 | 0.58 | 0.61 |
| CC (box) | 0.52 | 0.44 | 0.52 | 0.53 | 0.56 |
| CC (peaks) | 0.79 | 0.76 | 0.77 | 0.74 | 0.76 |
| CC (volume) | 0.83 | 0.88 | – | – | 0.83 |
| Mean CC for ligands | | | | | |

sodium pyrophosphate. The protein samples were washed twice with a wash solution containing 0.1% w/v trichloroacetic acid and 10 mM KH$_2$PO$_4$ and then resuspended in 1 M NaOH at 55 °C. The radioactivity was determined by liquid scintillation counting and compared with the specific radioactivity of the ATP to compute the amount of $^{32}$P incorporated in the protein samples. The protein concentration was determined by BCA assay kit (Thermo Fisher Scientific, cat# 23227).

## ATPase activity

The K$^+$ dependence of ATPase activity was determined in the presence of 120 mM NaCl and varied concentrations of KCl (0–20 mM). The Na$^+$ dependence of ATPase activity was determined in the presence of 20 mM KCl and varied concentrations of NaCl (0–120 mM). The amount of inorganic phosphate released from the reactions was measured by Enzchek™ phosphate assay kit (Thermo Fisher Scientific, cat# E6646). LMNG-solubilized wild-type (wt) and mutant Na$^+$/K$^+$-ATPase proteins were premixed with 1 mM ATP, 0.1 units of purine nucleoside phosphorylase (PNP) enzyme, 200 μM 2-amino-6-mercapto-7-methylpurine riboside (MESG) substrate and KCl or NaCl at different concentrations shown above. The reaction was initiated by adding 10 mM MgCl$_2$ and incubated at room temperate for 5 min. A negative control reaction was set up in the presence of 100 mM Ouabain. The absorbance at 360 nm was measured by Omega CLARIOstar Plus instrument (BMG Labtech). The protein concentration was determined by BCA assay kit (Thermo Fisher Scientific, cat# 23227).

## Electrophysiology

Patch-clamp methods were described previously[37,38]. All recordings were performed at a holding potential of 0 mV. The extracellular solution contained 120 mM NaOH, 4 mM MgCl$_2$, 0.5 mM EGTA, 15 mM TEA-OH, 7 mM NaCl or KCl, and 10 mM HEPES, set to pH 7.4 with aspartate. The cytoplasmic solution contained 20 mM NaCl, 125 mM aspartate, 0.5 mM EGTA, 0.5 mM MgCl$_2$, 0.2 mM CaCl$_2$, 20 mM HEPES, 1 mM KH$_2$PO$_4$, 6 mM Mg-ATP, 0.2 mM GTP, set to pH 7.4 with KOH.

Free Ca$^{2+}$ of cytoplasmic solutions was calculated to be 0.05 μM, and free Mg$^{2+}$ was calculated to be 0.4 mM using WEBMAX EXTENDED (http://www.stanford.edu/~cpatton/webmaxc/webmaxcE.htm). Highly polished dental wax coated pipette tips with inner diameters of 4-6 μm were employed generating a typical access resistance during recordings ranging from 1.2 to 4 MΩ. To activate Na$^+$/K$^+$-ATPase activity, solutions were maintained at 37 °C, a giga seals was established and the membrane was ruptured with mild suction. Cells were held at a holding potential of 0 mV and stabilized for greater than 30 s to allow for cytoplasmic diffusion of ATP other ionic constituents. After 30 s, the cells were rapidly switched among up to four parallel solution streams. The base extracellular solution containing an additional 7 mM NaCl and 0 mM KCl was rapidly exchanged with a solution containing 7 mM KCl and 0 additional NaCl. The presence of extracellular K$^+$ ions and cytoplasmic Na$^+$ with ATP allowed for the electrogenic cycle of pump activity to be measured as an outward current at 0 mV. All chemicals were of the highest grade available from Sigma-Aldrich. Capacitance, real-time current, resistance, and conductance recordings were performed using MATLAB and Capmeter V7 software[39].

### Cell surface protein expression

The amount of Na$^+$/K$^+$-ATPase expression on the cell surface was measured using the EZ-Link Sulfo-NHS-LC-Biotinylation Kit (Thermo Fisher, cat# 21435). Cultured HEK 293-F cells expressing wt or mutant Na$^+$/K$^+$-ATPase proteins were harvested and washed three times with ice-cold PBS (pH 8.0) to remove amine-containing components from culture medium. Cells were resuspended in PBS at a concentration of ~25 × 10$^6$ cells/ml, added 2 mM Sulfo-NHS-LC-Biotin reagent and incubated at room temperate for 30 min to label the surface proteins. 100 mM Tris pH 8.0 solution was added to quench the reactions. Cells were washed with PBS for three times and lysed in the lysis buffer containing 20 mM HEPES-Na pH 7.4, 250 mM NaCl, 10% (v/v) glycerol, 1% (w/v) LMNG. The biotinylated proteins were pulled down using Neutravidin agarose (Thermo Fisher, cat#29202). The biotinylated α3, β1, and FXYD6 were detected by SDS-PAGE and Western blot techniques using polyclonal antibodies against α3 (Abclonal, cat# A16036) (1:1000 dilution), β1 (Abclonal, cat# A12403) (1:1000 dilution), and FXYD6 (Abclonal, cat# A14339) (1:1000 dilution). Donkey HRP-conjugated anti-rabbit IgG antibody (Fisher Scientific, cat# AP182P) (1:10,000 dilution) was used as a secondary antibody. Actin served not only as a loading control (Supplementary Fig. 1h, WCL panel) but also as a cell surface labeling control (Supplementary Fig. 1h, IP panel) to make sure only the cell surface proteins were labeled. HRP-conjugated anti-actin antibody (Santa Cruz, cat# sc-47778-HRP) (1:2000 dilution) was used. Experiment was done in triplicate. Uncropped blots were included in the Source Data file.

### Statistical analysis and graph plotting

Unpaired $T$-tests (for 2-group comparison) and one-way ANOVA tests (for multiple-group comparison) were performed and graphs were plotted using Prism software (version 9.3.1, GraphPad). Data were represented as mean ± standard deviation (if applied).

### Reporting summary

Further information on research design is available in the Nature Research Reporting Summary linked to this article.

## Data availability

The data that support this study are available from the corresponding authors upon reasonable request. The cryo-EM maps of the human α3 Na$^+$/K$^+$-ATPase have been deposited in the Electron Microscope Data Bank (EMDB) under accession codes: EMD-27164 (Na$^+$-occluded state), EMD-27167 (K$^+$-occluded state), EMD-27168 (exoplasmic side-open state), EMD-27165 (cytoplasmic side-open state) and EMD-27166 (AMPPCP-bound cytoplasmic side-open state). The atomic coordinates for the human α3 Na$^+$/K$^+$-ATPase have been deposited to the RCSB Protein Data Bank (PDB) under accession codes: 8D3U (Na$^+$-occluded state), 8D3X (K$^+$-occluded state), 8D3Y (exoplasmic side-open state), 8D3V (cytoplasmic side-open state) and 8D3W (AMPPCP-bound cytoplasmic side-open state). Previously published structural data used from the PDB are listed below: 3WGU and 3B8E. The source data underlying Figs. 1c–e, 3g, h, 4i are provided as a Source Data file. Source data are provided with this paper.

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

## Acknowledgements

This work is dedicated to the memory of Prof. Beth Levine. Beyond her invaluable scientific support, without which this work would not be possible, her kindness, encouragement, and devotion will have an indelible impact for generations of scientists to come. We thank Dr. Gabriele Meloni for critical reading of the manuscript. Cryo-EM data were collected at the University of Texas Southwestern Medical Center Cryo-EM Facility, which is funded by the CPRIT Core Facility Support Award RP170644. We thank D Stoddard for technical support and facility access. This work was supported in part by the Howard Hughes Medical Institute (to B.L.), the NIH (grant R01GM136976 to X.-C.B), the Welch Foundation (grant I-1944 to X.-C.B) the Virginia Murchison Linthicum Scholar in Medical Research fund (to X.-C.B).

## Author contributions

P.T.N, X.-C.B., and B.L. conceived the project. P.T.N. optimized the protein expression and purification procedures and prepared samples for cryo-EM analyses. P.T.N. performed cell-surface protein expression, ATPase assays. T.S.T. and P.T.N. performed phosphorylation assays. C.D. and M.F. conducted patch-clamp whole-cell recordings. X.-C.B. and E.U. collected the cryo-EM data. X.-C.B and P.T.N carried out cryo-EM reconstruction and model building. P.T.N and X.-C.B. wrote the paper with inputs from other authors.

## Competing interests

The authors declare no competing interests.
