## [Peer Review File · Nature Communications]

Structural basis for gating mechanism of the human brain sodium-potassium pumpReviewers' Comments:

Reviewer #1:

Remarks to the Author:

In this manuscript, Nguyen and co-authors describe their structural and functional analyses of a human brain-specific isoform of the P2 Na⁺/K⁺-ATPase. They first discovered that the alpha3-beta1-FXYD6 complex can be overexpressed in the HEK cells and then performed their studies on this enzyme complex. They first demonstrated that the enzyme is located to the cell surface and functions as a Na⁺/K⁺ pump and then went on to determine five cryo-EM structures in various states of the ATPase functional cycle. Among the five observed structures, three resembled the previously reported states of the mammalian homologs: the Na⁺-occluded state, the K⁺-occluded state, and the exoplasmic side-open state. However, two structures were novel and corresponded to the cytoplasmic-open state that had not been determined previously. The capture of multiple functional states for the same enzyme has enabled these authors to propose an interesting dual-gating mechanism in which the ion-binding pocket – largely shared for Na⁺ and K⁺ ions – is either closed at both gates or open to one side only but never open to both sides. The authors further examined the biological effects of key residues involved in cytoplasmic gating (F90 and L93). Overall, the cryo-EM structural aspect of the work is well performed. Capturing the cytoplasmic-open state of the Na⁺/K⁺ pump is a significant achievement. This work represents an important contribution to the mechanistic understanding of the classic Na⁺/K⁺-ATPase.

The manuscript will benefit from a careful revision. we have two major issues.

First is numerous out-of-order figure citations. This problem may be solved by breaking up Figure 4 and insert the relevant panels in other figures.

The second major issue is about the wording used in their conclusion. "Our finding defines the gating mechanism". This is an overstatement. The author should consider replace "define" with "suggest". Quadruple mutation (4A) at the critical ion binding pocket was introduced to capture the cytoplasmic open state. While this is laudable, the observed structure may depart from the native state, given the substantial modifications. In fact, the authors should highlight this potential shortcoming in the Discussion section.

Below is a list of minor issues.

Line 89 and supplementary figure 1a, what is NTA? In the main text, three subunits are named as a3, b1 and FXYD6, while in Supplementary Figure 1a, it is A3, B1 and D6. A3 and a3 appear several times in the manuscript. It is confusing.

Line 94 and Fig. 1c-d, perhaps specify "4A" as "4A mutant"?

Line 113-118 and Fig. 1e-h, we cannot tell M7/M10 and M9 helices in Fig. 1e-h. Which one is M3, M4, M5? Please improve labeling.

Line 131, Q920 may not be a negatively charged residue?

Line 160, "BeF3⁻ (r.m.s.d. ~ 0.9) – "Å" is missing.

Line 169-170, "the A domain tilted ~ 7° along the vertical axis in the presence of BeF3⁻ relative to that of the K⁺-occluded state (Fig. 2e)". But in Fig. 2e it is labeled 7 Å. 7° or 7 Å?

Line 178-179, "The exoplasmic gate of the Na⁺/K⁺-ATPases is defined by the arrangement of the M4E relative to the M6 helix (Fig. 2e)." Please label M4E and M6 in Fig. 2e.

Line 188-189, "but maintained the same phosphorylation activity (Fig. 4g) compared to the WT": The V319A mutant does not seem to maintain the same phosphorylation activity compared to WT as shown in Fig. 4g.

Line 195-198, "With the exception of E324 (M4), most of the Na⁺ binding residues situated in the helices M5 (E773 and E776), M6 (D801 and D805), and M8 (Q920) only moved slightly within the Na⁺ binding pockets between the cytoplasmic side-open and the Na⁺-occluded state (Supplementary Fig. 8d)": It appears E324 moves slightly but not the five residues mentioned. And main text refers to E773 but Supplementary Figure 8d refers to N773. Please clarify. Also, it is difficult to see the labeled residues. Similar issue exists in Supplementary Figures 1 and 7.

Line 207, "Na⁺-occluded state, creating a large distance between M1 and M3 (Fig. 3e)." we cannot see M3 in Fig. 3e.

Line 223-224, "their helix axis toward the membranes while the M3 vertically shifts upward by ~3 Å (Supplementary Fig. 8b)." The ~3Å upward shift is not shown in the cited figure.

Line 238-239, "whereas in the exoplasmic side-open state, the exoplasmic gate is open while the cytoplasmic gate is closed (Supplementary Fig. 9)." Perhaps you are referring to Supplementary Fig. 10?

Line 276, "luminal membranes as a "sliding-door" mechanism in the SERCA (Fig. 9c) to expose the ion": Are you referring to Supplementary Figure 9c, not Fig. 9c?

Line 278-279, "(F90 and L93) distal from M3 (T286) creates a wide-open cytoplasmic gate for Na⁺ access (Fig. 3e; Fig. 4d - f)": Add the name of the protein here "in the human Na⁺/K⁺-ATPase"?

Line 518-520, the authors state that "The 3.5-Å resolution cryo-EM map of E1-ATP state is resolved in the highest quality with many well-defined side chains density for all of the α₃, β₁ and FX_{YD6}, which enabled de novo building of an accurate model": The clash score of this model as listed in Table 1 is too high (32.52) and should be improved.

Line 521, typo "with the help of the Na⁺-occludedstructure described above": occluded structure

Fig. 4 a-e, please define "back view" and "front view". Same in Supplementary Fig. 10

Table 1, B factors of Exoplasmic side-open 75/94. Is this a typo? And the clash score at 41.98 is too high and should be improved.

Supplementary Fig. 1 c,e: N773 or E773? Please clarify.

Supplementary Figs. 2 -6: Please show the FSC curves of model vs map and model vs half maps.

Supplementary Fig. 7b: Is this a superposition of the helices or a close-up view of the superposition of the overall structure in panel a? Please clarify. Same issue in Supplementary Fig. 8b.

Supplementary Fig. 11: Please show the EM densities for AMP-PCP, ALF4--ADP, MgF42- and BeF3-.

Reviewer #2:

Remarks to the Author:

This manuscript provides five cryo-em structures of the human neuron-specific alpha-3 isoform of the Na⁺,K⁺-ATPase. The work is very significant because:

- 1) It provides the first atomically-resolved structures of the alpha-3 isoform of the enzyme;
- 2) It provides the first ever structures of the cytoplasmic open states E1 and E1ATP of any isoform of the Na⁺,K⁺-ATPase, and
- 3) It provides the first structural detail of the cytoplasmic gating mechanism of the Na⁺,K⁺-ATPase, demonstrating in particular a large movement of the first transmembrane helix during the transition from the cytoplasmic open state to the exoplasmic open state.

The work supports the conclusions and claims. I can't detect any flaws in the data analysis, interpretation, conclusions or the methodology. The work definitely meets the expected standards in the field, and it contains enough detail for the work to be reproduced.

The work will be of significance to the field and related fields. The major criticisms I have are that the manuscript fails to cite some relevant publications in the established literature and that some points of detail regarding the structures presented are missing. All of my criticisms are listed below in order of their appearance in the manuscript, including some minor typographical points.

1. Summary, line 27

The authors state that their work reveals "an unprecedented" cytoplasmic gating mechanism of the Na⁺,K⁺-ATPase. The words "an unprecedented" should be replaced by "the first atomically-resolved structural detail of the". In fact there is a large body of biochemical data indicating that the transition of the enzyme from the E2 to the E1 state and the associated release of K⁺ to the cytoplasm and uptake of Na⁺ from the cytoplasm involves significant movement of the enzyme's lysine-rich extramembrane N-terminus on the Na⁺,K⁺-ATPase's cytoplasmic surface. This segment of the alpha-subunit is directly linked to the first transmembrane helix, which this work shows undergoes a large displacement during cytoplasmic gating. Thus, the manuscript provides strong support to the previous biochemical data. As such, the relevant biochemical data should be cited. This goes back to the measurement of tryptic digest patterns published by Jorgensen in 1975 (BBA 401 (1975) 399-415). This initial report by Jorgensen was later confirmed by subsequent studies and further details concerning the N-terminus movement were discovered. A short summary of the relevant findings can be found in the Introduction of the paper Jiang et al, Biophys J 112 (2017) 288-299.

2. Introduction, line 34

Please add "Christian" after "Jens". Skou was always referred to as Jens Christian, never just Jens.

3. Introduction, line 52

Change "an cytoplasmic" to "a cytoplasmic".

4. Introduction, line 68

Change "unknown" to "unclear". Because of the existence of the biochemical data (see point 1), "unknown" is too strong a word here.

5. Page 5, line 154

"in" appears twice. Remove one "in".

6. Page 17, lines 518-519

The authors state that the cytoplasmic domains were built based on homologous models using the published X-ray crystal structures of pig alpha-1 Na⁺,K⁺-ATPase PDB 3WGU and 3B8E. However, none of the published X-ray crystal structures have so far been able to resolve the complete Na⁺,K⁺-ATPase structure to the very start of the N-terminus. Therefore, does this mean that the N-termini are not resolved in these cryo-em structures either?

7. Page 17, line 525

The authors state that they were able to solve the entire complex of the E1ATP structure. Do they mean from the very start of the N-terminus of the alpha subunit? If this is true, then this would be the first completely resolved structure by any method.

8. Page 17, lines 526-528

The authors state that the P and N domains have poor densities because they possess high flexibility in the absence of ATP. This itself is interesting. In their rigid body fitting using the E1ATP structure, how much of the protein were they able to resolve. Were there still segments for which they couldn't determine a structure?

9. Fig. 1f

In the caption of this figure the authors state that "The black dashed line represents missing density in the structure for the amino acids." Which segment of the protein is this? The authors should include

the range of amino acid sequence numbers which are missing. This would allow an easier comparison with published X-ray crystal structures and the segments which are missing in those structures.

10. Figs. 3e, Suppl. Fig. 1b, Suppl. Fig. 7a, Suppl. Fig. 8a, Suppl. 11

Structures shown in these figures also have dashed black lines. Are these also segments with missing density? If that is correct, the authors should include for each structure precisely which amino acid sequence number ranges are missing in their structures.

Ronald J. Clarke
School of Chemistry, University of Sydney

Reviewer #3:

Remarks to the Author:

Nguyen et al report five cryo-EM structures of Na,K-ATPase alpha3-isoform in different intermediate states. In order to obtain inward- or outward-open states, the authors generate 4A mutant in which the cation-coordinating acidic side chains were mutated to alanine. The approach is unique, and this is the first report of an inward-open structure for the Na,K-ATPase. However, authors fall short in their functional analysis and its interpretation. The characteristics of neuron-specific alpha3 have hardly been discussed, and its relationship to pathology is not clear. Overall, the manuscript in the present form is immature for publication in Nature Communications. I hope following comments help improving the manuscript.

Major comments

Structural comparison

Throughout the manuscript, there is no description of how the structures are superimposed. As the authors themselves wrote, TM7-10 seems to function as an anchor. Especially when comparing cation-binding sites, structures should be superimposed by TM7-10. For the comparison of the cytoplasmic domains, on the other hand, P-domain may be an appropriate anchor. Thus, it was difficult to follow the differences in the figure because of the lack of information on how to superimpose them.

Functional assays

This is a serious problem. The superficial analysis of EP and ATPase activities of WT and mutant is not informative at all. Does all molecule accumulate in EP intermediate in the presence of 150 mM Na⁺? If one assumed 100% pure sample, the amount of 150 kDa protein complex in the sample is about 6.66 nmol/mg. Does the considerably low amount of EP (~1 nmol/mg) mean 5/6 molecules are inactive or denatured? Amount of EP accumulated is defined by the rate constant of Na-dependent EP formation and spontaneous EP dephosphorylation in the case of K⁺ absence. So that approx. 1 nmol/mg EP indicate that most of molecules are not accumulated in EP state in this condition. Different amount of EP is just because the relative rate constant for phosphorylation/dephosphorylation is changed in mutants.

Neither Na⁺ nor K⁺-dependence of ATPase activity are presented. Although authors judge the difference in ATPase activity between WT and F90A, L93A, V319A are significant, is it really V_{max} for all mutant at 130 mM Na and 20 mM K⁺? It is difficult to judge without having Na⁺- and K⁺-dependence. It was not written in the figure legend or methods what kind of data points are plotted. Are these multiplications using the same sample? Or are they values measured from independently expressed-purified sample? If the former, the error bars merely reflect the accuracy of the measurement technique. The specific activity is obtained by dividing the amount of protein determined by BCA assay. Therefore, the absolute value of ATPase activity is likely to vary more or less between purified samples. It is required to measure ATPase activity of independently prepared multiple samples to show their significant difference.

Pathophysiology of alpha3

As authors described, mutation of alpha3-isoform causes neurological diseases. However, there is no

description for the relationship between molecular structure and the diseases.

Overall, it is unclear what is unique about alpha3 isoform as a neuron-specific sodium pump. What discriminates alpha3 from housekeeping alpha1? This issue was not discussed at all.

Specific comments

L24 AMPPCP-bound cytoplasmic side-open (E1[dot]ATP). Throughout the manuscript and figures, description of reaction intermediates is not consistent.

L88 The combination of subunit used in this study is physiologically relevant?

L96 As wrote above, the amount of EP measured is unlikely to be the maximum value, and does not reflect the number of active sites in the sample. Compare the amount of 3H-ouabain bound to the sample if possible.

L104 In the present structure, it is unclear whether Na/K ions are actually occluded or not. This is an assumption based on the ligand added and the molecular structure obtained. It is desirable to give a correct description to avoid misunderstanding.

L108 Although overall resolution is 3.7Å, local resolution of TM helices looks better. In some cases cation is visible in 3~4Å resolution map, due to their positive charge and large atomic number (much less than X-ray, electron scattering factor also depends on the atomic number). Maybe Na is difficult, but K⁺ may visible even in low-resolution map. There are no EM density maps presented at the cation-binding site, and reader could not judge how models are reliable or not. The EM density maps of cation-binding site should be displayed in a form that includes amino acids and cation, not like a supplementary figure that only shows EM density around TM helices.

L117 Here authors themselves wrote TM7-10 act as an anchor. However, in some of the figures, structures are not superimposed by TM7-10. This is particularly useful for the comparison of the cation-binding site. For example, Fig S7 and S8, TM8-10 are arranged differently, which is inconsistent with the anchoring action of TM7-10. Specifically, in Fig S8, movement of Q920 in M8 during E1-ATP and (Na)E1P-ADP is unlikely. In addition, panel c and d in Fig S8 look differently (in c, TM5-10 is well overlapped, but these are shifted in similar degree in panel d). Same for Fig. S7.

L150 It is unclear the decreased activity of mutant is due to the reduction of apparent affinity or turnover number, by the measurement of fixed concentration of Na and K. At least, author should measure Na⁺ and K⁺-dependence to confirm the measured value is actually V_{max} or not.

L160 Experimental conditions for the determined structures were not described in the methods (high concentration of Na is applied for E1 state?), should be described in detail.

L194 this is the only difference regarding the conformational change between SERCA and NaK alpha3. Why does the movement differ between the two ATPases? The molecular mechanism is not described/discussed.

L256 ADP release does not induce TM4 movement. This is induced by the A domain rotation - TGES-P interaction - TM1-2 rearrangement, which is not described in this manuscript.

L283 As mentioned above, is it possible to discussion why SERCA and NKA alpha3 behave differently when the cytoplasmic gate is opened, in relation to A domain rotation and/or A-TM1 or TM2 linker. Many parts of discussion are redundant.

Comparison of the P-domain structure between E1, E1-ATP and E1P-ADP states are missing. It has been reported for SERCA E2-ACP state, D369 conformation is a key for the P domain bending (Kabashima et al, 2020, PNAS). In the case of SERCA, E1-AMPPCP and E1P-ADP states are

indistinguishable, which means ATPPCP-binding already mimics autophosphorylation. This is in marked contrast to this study (Inward-open for E1-ATP state, and Na-occluded for E1P-ADP state). So that P domain structure must be different between E1-ATP and E1P-ADP state of alpha3.

Response to questions/ comments of Reviewer #1

In this manuscript, Nguyen and co-authors describe their structural and functional analyses of a human brain-specific isoform of the P2 Na⁺/K⁺-ATPase. They first discovered that the alpha3-beta1-FXYD6 complex can be overexpressed in the HEK cells and then performed their studies on this enzyme complex. They first demonstrated that the enzyme is located to the cell surface and functions as a Na⁺/K⁺ pump and then went on to determine five cryo-EM structures in various states of the ATPase functional cycle. Among the five observed structures, three resembled the previously reported states of the mammalian homologs: the Na⁺-occluded state, the K⁺-occluded state, and the exoplasmic side-open state. However, two structures were novel and corresponded to the cytoplasmic-open state that had not been determined previously. The capture of multiple functional states for the same enzyme has enabled these authors to propose an interesting dual-gating mechanism in which the ion-binding pocket—largely shared for Na⁺ and K⁺ ions – is either closed at both gates or open to one side only but never open to both sides. The authors further examined the biological effects of key residues involved in cytoplasmic gating (F90 and L93). Overall, the cryo-EM structural aspect of the work is well performed. Capturing the cytoplasmic-open state of the Na⁺/K⁺ pump is a significant achievement. This work represents an important contribution to the mechanistic understanding of the classic Na⁺/K⁺-ATPase.

Thanks for the positive and constructive comments.

Below is the point-by-point response to your comments and questions,

1. First is numerous out-of-order figure citations. This problem may be solved by breaking up Figure 4 and insert the relevant panels in other figures.

Thank you for bringing this to our attention. We have reordered the main figures and supplementary figures substantially. We also break up the Figure 4 according to the suggestion.

2. The second major issue is about the wording used in their conclusion. “Our finding defines the gating mechanism”. This is an overstatement. The author should consider replace “define” with “suggest”. Quadruple mutation (4A) at the critical ion binding pocket was introduced to capture the cytoplasmic open state. While this is laudable, the observed structure may depart from the native state, given the substantial modifications. In fact, the authors should highlight this potential shortcoming in the Discussion section.

Thanks for raising this issue. We agree with the reviewer that we may have overstated the conclusion. In our revised manuscript, we have expanded our discussion and softened the tone of our conclusions. In particular, we have replaced “define” with “suggest” and have acknowledged the potential concern of introducing 4 mutations into the ion binding site in the revised Discussion.

Below is a list of minor issues.

3. Line 89 and supplementary figure 1a, what is NTA? In the main text, three subunits are named as a3, b1 and FXYD6, while in Supplementary Figure 1a, it is A3, B1 and D6. A3 and a3 appear several times in the manuscript. It is confusing.

We are sorry for causing the confusion. NKA was intended to represent the NKA or sodium potassium ATPase. To keep consistency, we have replaced “NKA” with “Na⁺/K⁺-ATPase”. We have changed A3, B1 and D6 to α 3, β 1 and FXYD6, respectively.

4. Line 94 and Fig. 1c-d, perhaps specify “4A” as “4A mutant”?

Point accepted. We have replaced “4A” with “4A mutant”.

5. Line 113-118 and Fig. 1e-h, we cannot tell M7/M10 and M9 helices in Fig. 1e-h. Which one is M3, M4, M5? Please improve labeling.

Good point. We have labeled the helices more clearly in the revised figure. We have also added additional figure panels (180°-rotated) to better show the FXYD6 and the helices from α subunit that are close to FXYD6. Please see Fig. 2 in the revised manuscript.

6. Line 131, Q920 may not be a negatively charged residue?

Good point. We have changed the wording to describe the cation-binding residues in the revised manuscript.

7. Line 160, “BeF3- (r.m.s.d. ~ 0.9)” – “Å” is missing.

Thank you for pointing this out. “Å” is added.

8. Line 169-170, “the A domain tilted ~ 7 \circ along the vertical axis in the presence of BeF3- relative to that of the K⁺-occluded state (Fig. 2e).” But in Fig. 2e it is labeled 7 Å. 7 \circ or 7 Å?

We have corrected the typo. It is 7°, not 7 Å.

9. Line 178-179, “The exoplasmic gate of the Na⁺/K⁺-ATPases is defined by the arrangement of the M4E relative to the M6 helix (Fig. 2e).” Please label M4E and M6 in Fig. 2e.

We have prepared additional panels with clear labelling of M4E and M6 to show the exoplasmic gate and illustrate the movement between M4E and M6 during gating. See Fig. 3d in the revised manuscript.

10. Line 188-189, “but maintained the same phosphorylation activity (Fig. 4g) compared to the WT”: The V319A mutant does not seem to maintain the same phosphorylation activity compared to WT as shown in Fig. 4g.

Thanks for raising this important point. The phosphorylation activity of the V319A mutant is slightly decreased compared to that of the WT ($p=0.1778$). We have changed the wording to reflect this minor change in the phosphorylation activity.

11. Line 195-198, “With the exception of E324 (M4), most of the Na⁺ binding residues situated in the helices M5 (E773 and E776), M6 (D801 and D805), and M8 (Q920) only moved slightly within the Na⁺ binding pockets between the cytoplasmic side-open and the Na⁺-occluded state (Supplementary Fig. 8d): It appears E324 moves slightly but not the five residues mentioned. And main text refers to E773 but Supplementary Figure 8d refers to N773. Please clarify. Also, it is difficult to see the labeled residues. Similar issue exists in Supplementary Figures 1 and 7.

Thanks for pointing out this mistake. We have corrected the typo. It is N773, not E773. We have also improved the clarity of the labeling. Please see the update supplementary Figs. 1 – 3 in the revised manuscript.

12. Line 207, “Na⁺-occluded state, creating a large distance between M1 and M3 (Fig. 3e).” we cannot see M3 in Fig. 3e.

We have renumbered Fig. 3 as Fig. 4 in the revised manuscript. In this figure, we have added Fig. 4f to better illustrate the large distance between M1 and M3 in the cytoplasmic side-open structure.

13. Line 223-224, “their helix axis toward the membranes while the M3 vertically shifts upward by ~3 Å (Supplementary Fig. 8b).” The ~3Å upward shift is not shown in the cited figure.

Good point. We have indicated the “~3Å upward shift” in the revised figure. Please see Supplementary Fig. 3b in the revised manuscript.

14. Line 238-239, “whereas in the exoplasmic side-open state, the exoplasmic gate is open while the cytoplasmic gate is closed (Supplementary Fig. 9).” Perhaps you are referring to Supplementary Fig. 10?

Thanks for pointing out the mistake. We have cited the correct figure in the revised manuscript.

15. Line 276, “luminal membranes as a “sliding-door” mechanism in the SERCA (Fig. 9c) to expose the ion”: Are you referring to Supplementary Figure 9c, not Fig. 9c?

Thanks for pointing out the mistake. We have cited the correct figure in the revised manuscript.

16. Line 278-279, “(F90 and L93) distal from M3 (T286) creates a wide-open cytoplasmic gate for Na⁺ access (Fig. 3e; Fig. 4d - f): Add the name of the protein here “in the human Na⁺/K⁺-ATPase”?

Good suggestion. We have added the protein name and revised the text to make a clearer point.

17. Line 518-520, the authors state that “The 3.5-Å resolution cryo-EM map of E1-ATP state is resolved in the highest quality with many well-defined side chains density for all of

the $\alpha 3$, $\beta 1$ and FXYD6, which enabled de novo building of an accurate model”: The clash score of this model as listed in Table 1 is too high (32.52) and should be improved.

We have re-refined the model with Phenix. The clash score is reduced to 9.56.

18. Line 521, typo “with the help of the Na⁺-occludedstructure described above”: occluded structure

We have corrected the typo.

19. Fig. 4 a-e, please define “back view” and “front view”. Same in Supplementary Fig. 10

Good suggestion. We have defined the “back view” and “front view” in the revised figure legend.

20. Table 1, B factors of Exoplasmic side-open 75/94. Is this a typo? And the clash score at 41.98 is too high and should be improved.

We have corrected the typo.

We have re-refined the model with Phenix. The clash score is reduced to 10.11.

21. Supplementary Fig. 1 c,e: N773 or E773? Please clarify.

As noted in an earlier correction, this was a typographical error and we thank the reviewer for bringing this to our attention. The proper notation should be N773 and it has been revised in the text accordingly.

22. Supplementary Figs. 2 -6: Please show the FSC curves of model vs map and model vs half maps.

We have shown the FSC curve between model and map/half maps in the revised figures.

23. Supplementary Fig. 7b: Is this a superposition of the helices or a close-up view of the superposition of the overall structure in panel a? Please clarify. Same issue in Supplementary Fig. 8b.

Sorry for the confusion. It is the close-up view of the superposition of the overall structure in panel a. We have added the description in the figure legend of both Supplementary Figs. Please see Supplementary Fig. 2 and 3 in the revised manuscript.

24. Supplementary Fig. 11: Please show the EM densities for AMP-PCP, AlF₄⁻-ADP, MgF₄²⁻ and BeF₃⁻.

Good suggestion. We have made additional figures in the revised manuscript to clearly show the EM densities for AlF₄⁻-ADP (Fig. 2c), MgF₄²⁻ (Fig. 2d), BeF₃⁻ (Supplementary Fig. 2a) and

AMPPCP (Supplementary Fig. 3e).

Response to questions/ comments of Reviewer #2

This manuscript provides five cryo-em structures of the human neuron-specific alpha-3 isoform of the Na⁺,K⁺-ATPase. The work is very significant because:

- 1) It provides the first atomically-resolved structures of the alpha-3 isoform of the enzyme;**
 - 2) It provides the first ever structures of the cytoplasmic open states E1 and E1ATP of any isoform of the Na⁺,K⁺-ATPase, and**
 - 3) It provides the first structural detail of the cytoplasmic gating mechanism of the Na⁺,K⁺-ATPase, demonstrating in particular a large movement of the first transmembrane helix during the transition from the cytoplasmic open state to the exoplasmic open state.**
- The work supports the conclusions and claims. I can't detect any flaws in the data analysis, interpretation, conclusions or the methodology. The work definitely meets the expected standards in the field, and it contains enough detail for the work to be reproduced. The work will be of significance to the field and related fields. The major criticisms I have are that the manuscript fails to cite some relevant publications in the established literature and that some points of detail regarding the structures presented are missing. All of my criticisms are listed below in order of their appearance in the manuscript, including some minor typographical points.**

Thanks for the positive and constructive comments.

1. Summary, line 27

The authors state that their work reveals "an unprecedented" cytoplasmic gating mechanism of the Na⁺,K⁺-ATPase. The words "an unprecedented" should be replaced by "the first atomically-resolved structural detail of the". In fact there is a large body of biochemical data indicating that the transition of the enzyme from the E2 to the E1 state and the associated release of K⁺ to the cytoplasm and uptake of Na⁺ from the cytoplasm involves significant movement of the enzyme's lysine-rich extramembrane N-terminus on the Na⁺,K⁺-ATPase's cytoplasmic surface. This segment of the alpha-subunit is directly linked to the first transmembrane helix, which this work shows undergoes a large displacement during cytoplasmic gating. Thus, the manuscript provides strong support to the previous biochemical data. As such, the relevant biochemical data should be cited. This goes back to the measurement of tryptic digest patterns published by Jorgensen in 1975 (BBA 401 (1975) 399-415). This initial report by Jorgensen was later confirmed by subsequent studies and further details concerning the N-terminus movement were discovered. A short summary of the relevant findings can be found in the Introduction of the paper Jiang et al, Biophys J 112 (2017) 288-299.

Point accepted. We have changed the words "an unprecedented" into "the first atomically-resolved structural detail". In addition, we would like to thank the reviewer for educating us about the history of the research on Na/K pump. We have cited the works published by Jorgensen et.al in the revised manuscript and fully appreciate the previous biochemical work that paved the way for our study.

2. Introduction, line 34

Please add "Christian" after "Jens". Skou was always referred to as Jens Christian, never

just Jens.

Thanks for pointing this out. We have revised the manuscript accordingly.

3. Introduction, line 52

Change "an cytoplasmic" to "a cytoplasmic".

Corrected

4. Introduction, line 68

Change "unknown" to "unclear". Because of the existence of the biochemical data (see point 1), "unknown" is too strong a word here.

Point accepted. We have changed “unknown” to “unclear”.

5. Page 5, line 154

"in" appears twice. Remove one "in".

Corrected

6. Page 17, lines 518-519

The authors state that the cytoplasmic domains were built based on homologous models using the published X-ray crystal structures of pig alpha-1 Na⁺,K⁺-ATPase PDB 3WGU and 3B8E. However, none of the published X-ray crystal structures have so far been able to resolve the complete Na⁺,K⁺-ATPase structure to the very start of the N-terminus. Therefore, does this mean that the N-termini are not resolved in these cryo-em structures either?

Similar to the previous crystal structures, the very N-termini of the α_3 subunit of human Na⁺/K⁺-ATPase was not resolved in our cryo-EM , indicating its structural flexibility. We have clarified this point in the revised manuscript.

7. Page 17, line 525

The authors state that they were able to solve the entire complex of the E1ATP structure. Do they mean from the very start of the N-terminus of the alpha subunit? If this is true, then this would be the first completely resolved structure by any method.

We apologize for the confusion caused. We meant to say that the E1ATP structure comprises all the three subunits. We have modified the text and removed the term “the entire complex”.

8. Page 17, lines 526-528

The authors state that the P and N domains have poor densities because they possess high flexibility in the absence of ATP. This itself is interesting. In their rigid body fitting using the E1ATP structure, how much of the protein were they able to resolve. Were there still segments for which they couldn't determine a structure?

Despite the weak densities of the P and N domains, we could build a near-complete structure for these soluble regions by rigid-body fitting each domain from previous crystal structures. Only some loop regions were poorly resolved in cryo-EM map. There were no segments for which we couldn't determine the structure.

9. Fig. 1f

In the caption of this figure the authors state that "The black dashed line represents missing density in the structure for the amino acids." Which segment of the protein is this? The authors should include the range of amino acid sequence numbers which are missing. This would allow an easier comparison with published X-ray crystal structures and the segments which are missing in those structures.

Good suggestion. We have indicated the residues numbers which are missing in the structural models in the figure legends.

10. Figs. 3e, Suppl. Fig. 1b, Suppl. Fig. 7a, Suppl. Fig. 8a, Suppl. 11

Structures shown in these figures also have dashed black lines. Are these also segments with missing density? If that is correct, the authors should include for each structure precisely which amino acid sequence number ranges are missing in their structures.

Good suggestion. We have indicated the range of amino acid sequence numbers that are missing in our structural models.

Response to questions/ comments of Reviewer #3

Nguyen et al report five cryo-EM structures of Na,K-ATPase alpha3-isoform in different intermediate states. In order to obtain inward- or outward-open states, the authors generate 4A mutant in which the cation-coordinating acidic side chains were mutated to alanine. The approach is unique, and this is the first report of an inward-open structure for the Na,K-ATPase. However, authors fall short in their functional analysis and its interpretation. The characteristics of neuron-specific alpha3 have hardly been discussed, and its relationship to pathology is not clear. Overall, the manuscript in the present form is immature for publication in Nature Communications. I hope following comments help improving the manuscript.

We are grateful for the feedbacks and constructive suggestions from this reviewer.

1. Structural comparison

Throughout the manuscript, there is no description of how the structures are superimposed. As the authors themselves wrote, TM7-10 seems to function as an anchor. Especially when comparing cation-binding sites, structures should be superimposed by TM7-10. For the comparison of the cytoplasmic domains, on the other hand, P-domain may be an appropriate anchor. Thus, it was difficult to follow the differences in the figure because of the lack of information on how to superimpose them.

We apologize for the lack of description of how the structures are superimposed. We have described how the structures were aligned in all of the figure legends and texts where superposition of structures were done in the revised manuscript.

While in supplementary Fig. 7e and 8e, we superimposed the structures by the TM7 – 10 to show the movement of the M1 – M6 helices; we made a mistake in superposition of the cation-binding sites (Supplementary Fig. 7f and 8f). We have updated our superimposition by aligning the M7 - 10 helices to compare cation-binding sites in different structural states in the Supplementary Fig. 2f and 3c in the revised manuscript.

In addition, we have also added superpositions of the cytoplasmic domains by the P domain to show the rearrangement of the A, P and N domains in different states. See Supplementary Fig. 3f and Supplementary Fig. 7b in the revised manuscript.

2. Functional assays

This is a serious problem. The superficial analysis of EP and ATPase activities of WT and mutant is not informative at all. Does all molecule accumulate in EP intermediate in the presence of 150 mM Na⁺? If one assumed 100% pure sample, the amount of 150 kDa protein complex in the sample is about 6.66 nmol/mg. Does the considerably low amount of EP (~1 nmol/mg) mean 5/6 molecules are inactive or denatured? Amount of EP accumulated is defined by the rate constant of Na-dependent EP formation and spontaneous EP dephosphorylation in the case of K⁺ absence. So that approx. 1 nmol/mg EP indicate that most of molecules are not accumulated in EP state in this condition. Different amount of EP

is just because the relative rate constant for phosphorylation/dephosphorylation is changed in mutants.

Thanks for raising these concerns. We agree with the reviewer that there should be ~6.66 nmol phosphorylation sites per 1 mg protein (~6.66 nmol $\alpha 3$ subunit). However, it is worth noting that to achieve full-activity of Na/K ATPase in the in-vitro experiments requires the fulfillment of several factors, including optimal pH and temperature (7.4 and 37 degree), no dephosphorylation activity, 100% sample purify and the presence of lipid environment. In practice, the phosphorylation reaction was performed at 0 °C in a short time (20 seconds) to reduce the dephosphorylation activity (Lupfert et. al., *Biophysical journal* 81, 2069 – 2081, 2001). In these suboptimal conditions, we were unable to achieve the theoretical 6.66 nmol phosphorylation site per 1 mg protein. In consistent with our results, Lupfert et. al. (*Biophysical Journal* 81, 2069 – 2081, 2001) also showed that the number of phosphorylation sites per 1 mg mammalian $\alpha 1$ Na⁺/K⁺-ATPase proteins measured was suboptimal, i.e., ~2.43 sites/mg of protein from pig kidney and ~3.24 sites/mg of protein from rabbit kidney.

In this manuscript, we purposely measured and compared the relative enzymatic activity of different variants of the human $\alpha 3$ Na⁺/K⁺-ATPase (WT versus selected mutants) in the same set-up conditions to demonstrate the role of key gating residues at the cytoplasmic and exoplasmic gates. In the revised manuscript, we also included SDS-page results to demonstrate that the amount and purify of protein used ATPase activity assay was similar between WT and mutant proteins. A relative activity comparison using the WT phosphorylation and ATPase activities of the WT as 100% has been done by other publications, e.g. Morth et. al., *Nature* 459, 13, 1043 – 1050, 2007.

3. Neither Na⁺ nor K⁺-dependence of ATPase activity are presented. Although authors judge the difference in ATPase activity between WT and F90A, L93A, V319A are significant, is it really V_{max} for all mutant at 130 mM Na and 20 mM K⁺? It is difficult to judge without having Na⁺- and K⁺-dependence.

Thank the reviewer for the good suggestion. We have performed the K⁺ and Na⁺-dependent ATPase activity, shown in Fig. 3h and 4i, respectively in the revised manuscript. The new data indicated that the V_{max} for the WT and all mutants of the human Na⁺/K⁺-ATPase could be obtained in the buffer containing 120 mM Na⁺ and 20 mM K⁺.

4. It was not written in the figure legend or methods what kind of data points are plotted. Are these multiplications using the same sample? Or are they values measured from independently expressed-purified sample? If the former, the error bars merely reflect the accuracy of the measurement technique. The specific activity is obtained by dividing the amount of protein determined by BCA assay. Therefore, the absolute value of ATPase activity is likely to vary more or less between purified samples. It is required to measure ATPase activity of independently prepared multiple samples to show their significant difference.

Thanks for pointing out this issue. We have detailed the data acquisition method in the “Method” section. In brief, we measured ATPase activity of 3 different batches of purified protein samples in duplicate (n=6). We recorded the absorbance at 5 minute time point to calculate the ATPase

activity since we observed the Pi product released was not in plateau within 5 minutes of the reaction.

We have shown the SDS-PAGE data of all the recombinantly purified WT and mutants to show their purify (see **Supplementary Fig. 1b** in the revised manuscript). The accurate amount of protein loaded in the activity assay was determined by BCA assay.

4. Pathophysiology of alpha3

As authors described, mutation of alpha3-isoform causes neurological diseases. However, there is no description for the relationship between molecular structure and the diseases.

Thank the review for this comment. Our work has provided the first experiment-derived structures of the human $\alpha 3$ Na^+/K^+ -ATPase for mapping of the neurological disease related mutations. At the resolution of $\sim 3 - 4 \text{ \AA}$, however, we were unable to confidently detect small conformational change caused by the disease-related mutations.

Notably, in this work, we utilized a recombinant system for expression and purification of the exogenous human Na^+/K^+ -ATPase. The importance of this novel procedure lies in creating new opportunities for genetic manipulation of the $\alpha 3$ ATPase. This will allow us and others to study structure and function of the Na^+/K^+ -ATPase with disease related mutations introduced in the future. Therefore, our work provides new opportunities to understand the structure-function relationship and pathophysiology of the human $\alpha 3$ ATPase subunit.

5. Overall, it is unclear what is unique about alpha3 isoform as a neuron-specific sodium pump. What discriminates alpha3 from housekeeping alpha1? This issue was not discussed at all.

Thank the reviewer for the comment. We compared the overall structure between the human $\alpha 3$ and the pig $\alpha 1$ Na^+/K^+ -ATPase in our manuscript in the “Result” session. We showed that the two isoforms share similar overall structures in the Na^+ -occluded, K^+ -occluded and exoplasmic side-open states, although there are some minor structural differences. We have added more discussion in the revised manuscript.

5. L24 AMPPCP-bound cytoplasmic side-open (E1[dot]ATP). Throughout the manuscript and figures, description of reaction intermediates is not consistent.

Thank the reviewer for pointing out the inconsistent description. We have fixed this issue. The AMPPCP-bound cytoplasmic side-open state is now described as E1•ATP.

6. L88 The combination of subunit used in this study is physiologically relevant?

According to the human protein atlas database (<https://www.proteinatlas.org/>), the $\alpha 3$, $\beta 1$, FXVD6 were detected in different parts of the human brain. Moreover, the co-expression of $\alpha 3$, $\beta 1$, FXVD6 resulted in the highest expression level of Na/K ATPase, These two evidences

support that the combination of the $\alpha 3/\beta 1/\text{FX} \text{YD}6$ used in our structural study is physiologically relevant.

7. L104 In the present structure, it is unclear whether Na/K ions are actually occluded or not. This is an assumption based on the ligand added and the molecular structure obtained. It is desirable to give a correct description to avoid misunderstanding.

Good suggestion. We have changed the wording to avoid the misunderstanding.

8. L108 Although overall resolution is 3.7Å, local resolution of TM helices looks better. In some cases cation is visible in 3~4Å resolution map, due to their positive charge and large atomic number (much less than X-ray, electron scattering factor also depends on the atomic number). Maybe Na is difficult, but K⁺ may visible even in low-resolution map. There are no EM density maps presented at the cation-binding site, and reader could not judge how models are reliable or not. The EM density maps of cation-binding site should be displayed in a form that includes amino acids and cation, not like a supplementary figure that only shows EM density around TM helices.

Thank the reviewer for the suggestion. We have included figure panels (Fig. 2g and h in the revised manuscript) showing densities of the cation-binding sites and described more details in the “Results” section.

9. L117 Here authors themselves wrote TM7-10 act as an anchor. However, in some of the figures, structures are not superimposed by TM7-10. This is particularly useful for the comparison of the cation-binding site. For example, Fig S7 and S8, TM8-10 are arranged differently, which is inconsistent with the anchoring action of TM7-10. Specifically, in Fig S8, movement of Q920 in M8 during E1-ATP and (Na)E1P-ADP is unlikely. In addition, panel c and d in Fig S8 look differently (in c, TM5-10 is well overlapped, but these are shifted in similar degree in panel d). Same for Fig. S7.

Thank the review for pointing out this issue.

We have re-performed all the structural superimpositions by aligning the TM7-10 in order to illustrate the movement of TM1-6 and compare cation-binding sites in different structural states in the revised manuscript. See Supplementary Fig. 2g and Supplementary Fig. 3d in the revised manuscript.

10. L150 It is unclear the decreased activity of mutant is due to the reduction of apparent affinity or turnover number, by the measurement of fixed concentration of Na and K. At least, author should measure Na⁺ and K⁺-dependence to confirm the measured value is actually V_{max} or not.

Thank the reviewer for the suggestion. We have performed the K⁺ and Na⁺-dependent ATPase activity. The data have been added to Fig. 3h and 4i in the revised manuscript. In brief, the K⁺-dependent ATPase data suggested that the V319A mutation does not change its apparent K_m toward K⁺ but significantly increases its ATP turnover rate compared to the WT. The Na⁺-

dependent ATPase data suggested that the F90A and L93A mutation increase the apparent K_m and slightly reduce ATP hydrolysis activity. The data also indicated that the V_{max} for the WT and all mutants of the human Na^+/K^+ -ATPase could be obtained in the buffer containing 130 mM Na^+ and 20 mM K^+ .

11. L160 Experimental conditions for the determined structures were not described in the methods (high concentration of Na is applied for E1 state?), should be described in detail.

We have elaborated the experimental conditions for the structural determination of the Na^+/K^+ -ATPase at different states in the “sample preparation and EM data acquisition” (“Methods” section) in our revised manuscript.

12. L194 this is the only difference regarding the conformational change between SERCA and NaK alpha3. Why does the movement differ between the two ATPases? The molecular mechanism is not described/discussed.

We speculate that the different conformational changes of the M1 helices between these two type of ATPases is as a result of the different movement of the A domains. While the SERCA’s A domain moves up ~ 12 Å along a vertical axis (perpendicular to the lipid bilayers) (Supplementary Fig. 5c), the Na^+/K^+ -ATPase’s M1 helix rotates outward relative to the P domain between the E1 and E1•ATP structures. We have added more details in the “Discussion” section.

13. L256 ADP release does not induce TM4 movement. This is induced by the A domain rotation - TGES-P interaction – TM1-2 rearrangement, which is not described in this manuscript.

Good point. We have added a supplementary figure (Supplementary Fig. 7) to show the rotation of the A domain and the TGES motif that leads to the exoplasmic side-open.

14. L283 As mentioned above, is it possible to discussion why SERCA and NKA alpha3 behave differently when the cytoplasmic gate is opened, in relation to A domain rotation and/or A-TM1 or TM2 linker. Many parts of discussion are redundant.

Thank the reviewer for the suggestion. We have added more discussion of the different rotations of the M1 helix between the two ATPase in the revised manuscript. In brief, we speculate that these different operations of the two M1 helices are due to the different movements of the A domains that directly link to the M1 helices of the two ATPases. More specifically, while the SERCA’s A domain moves up ~ 12 Å along a vertical axis (perpendicular to the lipid bilayers) (Supplementary Fig. 5c) pushing the M1 toward the lumen membrane, the Na^+/K^+ -ATPase’s A domain rotates outward relative to the P domain in the E1 and E1•ATP structures compared to that in the E1•P-ADP structure.

Note that while the SERCA is made of a single alpha subunit, the Na^+/K^+ -ATPases are made of multiple subunits, including alpha, beta and FXYD. We can’t rule out the possibility that the beta and FXYD may play a certain roles in regulating the cytoplasmic gate of the Na^+/K^+ -ATPases.

Hence, further functional studies need to be done to understand any potential role(s) of the beta and FXYD subunits in the cytoplasmic gating mechanism of the Na⁺/K⁺-ATPases.

15. Comparison of the P-domain structure between E1, E1-ATP and E1P-ADP states are missing. It has been reported for SERCA E2-ACP state, D369 conformation is a key for the P domain bending (Kabashima et al, 2020, PNAS). In the case of SERCA, E1-AMPPCP and E1P-ADP states are indistinguishable, which means ATPPCP-binding already mimics autophosphorylation. This is in marked contrast to this study (Inward-open for E1-ATP state, and Na-occluded for E1P-ADP state). So that P domain structure must be different between E1-ATP and E1P-ADP state of alpha3.

Thank the reviewer for raising this good point. We have added a supplementary figure (Supplementary Fig. 3f) to show the superposition of the cytoplasmic domains of the AMPPCP-bound cytoplasmic side-open (E1•ATP) and Na⁺-occluded (E1•P-ADP) states by the P domain. While the P domain structure is similar in both the cytoplasmic side-open (E1) and AMPPCP-bound cytoplasmic side-open (E1•ATP), it shows different structural arrangement in the Na⁺-occluded state (E1•P-ADP). In particular, the helix 708-717 rotates about ~ 14° closer to the N domain in the E1•P-ADP compared to that in E1•ATP state. This rotation creates a hydrogen bond between residue N710 and the gamma phosphate group of the ATP molecule which may facilitate the autophosphorylation reaction.

Reviewers' Comments:

Reviewer #1:

Remarks to the Author:

Our concerns have been addressed satisfactorily in this revised manuscript.

Three minor changes:

1. Supplementary Figures 2g and 3d: In Figure captions, it should be clearly stated that the N773/E776/D801/D805 were modeled for the purpose of comparison to illustrate the side chain movement between states; they were not resolved experimentally because they were mutated to 4 alanines.

2. Supplementary Figures 8-12: Please provide rationale for using 0.4 rather than the standard 0.5 cutoff FSC between model and map.

3. A typo: 2.17+/- 0.27 pA/p -- pA/pF

Reviewer #2:

Remarks to the Author:

The authors have responded appropriately to all of my comments in their "Response to Reviewers" document.

The only criticism I have is that in the revised version of the manuscript the only additions in red are figure references. None of the changes that they made to address my comments are highlighted in red. This makes it very difficult to find them and unnecessarily wastes my time.

Reviewer #3:

Remarks to the Author:

The manuscript improves in structural comparison and its understanding, authors do not sufficiently respond to my previous comments, especially those for functional analysis and its interpretation.

Authors replied that they performed **duplicate** experiment of **three different batches** in their response. But looking at raw data, these are obviously **triplicate** of **two different batches**, and most of them are not well reproduced. I will show some examples below.

In the raw data table, KCl dependence and NaCl dependence of WT are ...

KCl (mM)	WT					
0	128	94	106	27	0	4
0.05	114	281	128	59	43	175
0.1	99	94	96	186	24	30
0.2	128	111	128	202	161	157
0.5	184	204	202	202	175	187
1	270	289	277	277	251	255
2	355	417	341	335	336	302
5	369	409	394	367	331	315
10	383	409	383	346	312	306
20	355	400	404	372	336	311
30	341	434	404	415	374	353

NaCl (mM)	WT					
0	281	286	267	153	170	159
1	255	259	250	170	163	165
2	272	286	261	255	198	204
5	366	443	335	221	232	210
10	332	341	329	272	286	278
20	375	361	346	366	381	369
40	349	361	346	417	429	431
60	375	361	346	434	456	426
80	341	354	341	375	395	380
100	332	334	329	375	388	386
120	375	395	369	426	443	426

Especially in the absence of K⁺ or Na⁺ and their low concentration range, the data differs

significantly between former three data and latter three.

If I plot likely two different triplicate measurement with Mean +/- SD, the data look like this...

However, to my surprise, authors average these all data, and underestimate errors as calculated in SEM, not in SD. Na⁺-dependence of ATPase activity starts from 150 or 280 μmol/mg/h in the absence of Na⁺, which is unlikely for WT NaKATPase activity. Contamination of certain amount of Na⁺ can be expected. Ouabain may be much suitable and reliable blank rather than -Mg, because only trace amount of Mg can proceed phosphorylation, and in some of old papers excess amount (~10 mM) of EDTA or CDTA was usually added to remove free divalent cations from the reaction.

Situation is much serious for L93A

KCl dependence

L93A						
1	156	102	106	11	5	9
1	71	119	74	21	9	21
3	85	119	64	21	28	21
3	14	187	11	32	19	21
5	28	289	11	32	28	34
3	14	409	32	53	52	43
1	128	570	117	106	104	98
5	184	562	181	197	184	187
5	241	536	245	255	251	247
5	284	494	287	293	279	281
0	284	443	213	277	260	264

NaCl dependence

	T	U	V	W	X	Y
	L93A					
2	136	136	142	9	27	11
1	128	136	125	128	123	62
2	128	129	136	136	123	57
1	153	170	165	179	163	102
9	204	204	210	213	198	131
5	204	218	233	272	279	204
5	238	245	250	306	320	233
1	315	259	278	349	354	267
6	272	279	261	392	361	301
1	281	272	289	409	388	306
8	272	266	272	358	354	289

I cannot trust the K_m and V_{max} values and statistical significance obtained from such a scattered and non-reproducible data. Clearly, data and analysis do not reach to the level required for the Nat Commun in the present form.

Interpretations of the functional data are also biased. Although scattered, V319A showed higher affinity for K^+ . Author's interpretation is that smaller side chain alanine increases the accessibility of K^+ to the cation binding site, and as a consequence apparent affinity for K^+ is increased. However, this interpretation does not explain its lower affinity for Na^+ . It is more likely that this mutation shifts its E1-E2 equilibrium toward E2 that favors K^+ .

Interpretations for F90A and L93A are also not correctively written. These two mutations reduce apparent Na^+ affinity relative to WT, if the affinity measurement is true. If smaller side chain works similarly to the V319A, accessibility of Na^+ to the cation binding site might be increased. As already known for SERCA, F90 and L93 are important as backing support for E324 or M4E in the Na^+ occluded state, and their replacement to smaller alanine side chain might weaken the occluded state and apparent affinity is reduced. However, these all mechanistic interpretations are on the basis of Na^+ or K^+ dependent ATPase measurement,

and in this paper their measurement is clearly insufficient to support above-mentioned interpretations.

As this reviewer wrote in the previous review comment, the amount of EP accumulated is determined by the relative rate constant of Na-dependent phosphorylation / K⁺-dependent dephosphorylation. The amount of EP accumulated is not “activity”. If the amount of EP for some mutant is lower than WT, it does not indicate that mutant has lower activity. For example, the amount of EP in F90A mutant is lower than WT. Because E2P dephosphorylation in the absence of K⁺ is one of the rate-limiting step, ATPase activity in the absence of K⁺ can be assumed to be the rate of spontaneous E2P dephosphorylation. In this case, average value of WT (59.8; average of 128, 94, 106, 27, 0, 4, from raw data table) and F90A (57.3; average of 99, 77, 96, 32, 19, 21) are similar (too scattered though), the reduced amount of EP in the F90A mutant is likely due to the slower rate of Na⁺-dependent phosphorylation for F90A relative to WT. Such a simple logic that requires to lead a conclusion has not been described at all in the manuscript. I recommend authors to ask some NaK-ATPase specialist to improve interpretations of all functional data, when they submit this paper elsewhere.

Another serious problem is that authors do not address what makes neuron-specific NaK different from alpha1. In the response, authors wrote they describe about it in the results. But I cannot find such description or any discussion related to this issue. In the abstract, the authors emphasize the fact that they have analyzed the structure of neuron-specific NaK, but they end up arguing that it is almost identical to alpha1 and do not discuss the alpha3-specific aspect at all. If authors fail to find any difference, then it is not important enough as a new discovery and has not reached the level required for this journal.

L281~

No exception, for the all the active transport proteins, “alternating access” principle is mandatory. It has been proposed long time ago, for example by Peter Mitchell (1957 Nature, 180, 134) and reviewed by David (Gadsby DC, 2009, Nat Rev Mol Cell Biol.). This is not the first time that this has been clarified in this paper, and there is no need to emphasize it as a “dual gating model”.

Fig.1c

This panel was not found in the previous version. Either bac-mam or PEI transfection using HEK cells are now utilized for many structural and functional analysis of mammalian

membrane proteins. For P-type ATPase, bac-mam is first applied for HK-ATPase, and also for many P4- and P5-ATPases. This expression strategy is not new at all, authors just applied this technique for alpha3 isoform.

Response to questions/ comments of Reviewer #1

Our concerns have been addressed satisfactorily in this revised manuscript.

We are very thankful for reviewer #1's support and approval for our revised manuscript. Below are the point-by-point responses to reviewer #1's remaining questions and suggestions:

1. Supplementary Figures 2g and 3d: In Figure captions, it should be cleared stated that the N773/E776/D801/D805 were modeled for the purpose of comparison to illustrate the side chain movement between states; they were not resolved experimentally because they were mutated to 4 alanines.

Point accepted. Thank you for your suggestion.

2. Supplementary Figures 8-12: Please provide rationale for using 0.4 rather than the standard 0.5 cutoff FSC between model and map.

Thanks for raising this issue. There are lots of discussion about the best cutoff for FSC between model and map. In theory, the cutoff for the FSC between model and map should be the square root of the cutoff for FSC between two half maps. Therefore, if one uses 0.143 for the map-to-map FSC, one should use 0.378 (~0.4) for the model-to-map FSC. One example could be found here (PMID: 26829225).

3. A typo: 2.17+/- 0.27 pA/p -- pA/pF

Thank you. We have fixed the typo.

Response to questions/ comments of Reviewer #2

We are very thankful for reviewer #2's approval for our revision. We sincerely apologize for not having highlighted the changes in our previous revised manuscript. We have highlighted these changes (in red) in this second revision.

Response to questions/ comments of Reviewer #3

The manuscript improves in structural comparison and its understanding, authors do not sufficiently respond to my previous comments, especially those for functional analysis and its interpretation.

We would like to thank this reviewer for the valued feedbacks. We have considered this reviewer #3 suggestion very carefully and incorporated ouabain as a negative control in our new functional assays. Our new functional data are highly reproducible with small standard deviations among 6 individual replicates (see below).

Below are the point-by-point responses to this reviewer's remaining questions and suggestions:

1. Authors replied that they performed duplicate experiment of three different batches in their response. But looking at raw data, these are obviously triplicate of two different batches, and most of them are not well reproduced. I will show some examples below.

We apologized for the typo in the previous point-by-point response letter. We performed the functional data in triplicate using two different batches of the purified proteins. We described this experimental setup in our Methods section in the previous revised manuscript (line 636-637).

2. In the raw data table, KCl dependence and NaCl dependence of WT are ...

(I attached pdf in which some figures and tables are included)

Especially in the absence of K⁺ or Na⁺ and their low concentration range, the data differs significantly between former three data and latter three.

If I plot likely two different triplicate measurement with Mean +/- SD, the data look like this...

However, to my surprise, authors average these all data, and underestimate errors as calculated in SEM, not in SD. Na⁺-dependence of ATPase activity starts from 150 or 280 $\mu\text{mol}/\text{mg}/\text{h}$ in the absence of Na⁺, which is unlikely for WT NaKATPase activity. Contamination of certain amount of Na⁺ can be expected. Ouabain may be much suitable and reliable blank rather than -Mg, because only trace amount of Mg can proceed phosphorylation, and in some of old papers excess amount (~10 mM) of EDTA or CDTA was usually added to remove free divalent cations from the reaction.

Situation is much serious for L93A

I cannot trust the Km and Vmax values and statistical significance obtained from such a scattered and non-reproducible data. Clearly, data and analysis do not reach to the level required for the Nat Commun in the present form.

Thank you so much for re-analyzing our previous functional data and giving us great suggestions, including the use of ouabain as a negative control. We have incorporated ouabain a negative control in our new functional assays. Indeed, the non-specific ATPase activity has been significantly diminished. Our new functional data are highly reproducible with small standard deviations among 6 individual replicates (see below).

3. Interpretations of the functional data are also biased. Although scattered, V319A showed higher affinity for K^+ . Author's interpretation is that smaller side chain alanine increases the accessibility of K^+ to the cation binding site, and as a consequence apparent affinity for K^+ is increased. However, this interpretation does not explain its lower affinity for Na^+ . It is more likely that this mutation shifts its E1-E2 equilibrium toward E2 that favors K^+ .

Thank you so much for your comments and suggestions. We have considered your comments carefully and incorporated them into our interpretations of our new functional data.

4. Interpretations for F90A and L93A are also not correctively written. These two mutations reduce apparent Na^+ affinity relative to WT, if the affinity measurement is true. If smaller side chain works similarly to the V319A, accessibility of Na^+ to the cation binding site might be increased. As already known for SERCA, F90 and L93 are important as backing support for E324 or M4E in the Na^+ occluded state, and their replacement to smaller alanine side chain might weaken the occluded state and apparent affinity is reduced. However, these all mechanistic interpretations are on the basis of Na^+ or K^+ dependent ATPase measurement, and in this paper their measurement is clearly insufficient to support above-mentioned interpretations.

As this reviewer wrote in the previous review comment, the amount of EP accumulated is determined by the relative rate constant of Na^+ -dependent phosphorylation / K^+ -dependent dephosphorylation. The amount of EP accumulated is not "activity". If the amount of EP for some mutant is lower than WT, it does not indicate that mutant has lower activity. For example, the amount of EP in F90A mutant is lower than WT. Because E2P dephosphorylation in the absence of K^+ is one of the rate-limiting step, ATPase activity in the absence of K^+ can be assumed to be the rate of spontaneous E2P dephosphorylation. In this case, average value of WT (59.8; average of 128, 94, 106, 27, 0, 4, from raw data table) and F90A (57.3; average of 99, 77, 96, 32, 19, 21) are similar (too scattered though), the reduced amount of EP in the F90A mutant is likely due to the slower rate of Na^+ -dependent

phosphorylation for F90A relative to WT. Such a simple logic that requires to lead a conclusion has not been described at all in the manuscript. I recommend authors to ask some NaK-ATPase specialist to improve interpretations of all functional data, when they submit this paper elsewhere.

Thank you so much for your comments and suggestions. Using ouabain as a negative control in our new functional assays, we have greatly minimized non-specific ATPase activities. We have also incorporated your comments/suggestions into our interpretations of our new functional data.

5. Another serious problem is that authors do not address what makes neuron-specific NaK different from alpha1. In the response, authors wrote they describe about it in the results. But I cannot find such description or any discussion related to this issue. In the abstract, the authors emphasize the fact that they have analyzed the structure of neuron-specific NaK, but they end up arguing that it is almost identical to alpha1 and do not discuss the alpha3-specific aspect at all. If authors fail to find any difference, then it is not important enough as a new discovery and has not reached the level required for this journal.

(last-revision question) Pathophysiology of alpha3 As authors described, mutation of alpha3-isoform causes neurological diseases. However, there is no description for the relationship between molecular structure and the diseases.

(last-revision question) Overall, it is unclear what is unique about alpha3 isoform as a neuron-specific sodium pump. What discriminates alpha3 from housekeeping alpha1? This issue was not discussed at all.

We apologize if the introduction of the neuron-specific alpha3 and its neurological disease mutations misled the ultimate goals of our manuscript.

In this work, we focus on characterizing structural details of a human Na⁺/K⁺-ATPase isoform in different states and revealing the structural basis for the gating mechanism of the Na⁺/K⁺-ATPase. Therefore, in this revision, we have removed the introduction of the neuron-specific alpha3 isoform and its neurological disease-related mutations in the introduction.

Our work has revealed the first atomically-resolved structural detail of the cytoplasmic gating mechanism of the Na⁺/K⁺-ATPase. We believed this new discovery will advance structural understanding of the Na⁺/K⁺-ATPase.

6. L281~

No exception, for the all the active transport proteins, “alternating access” principle is mandatory. It has been proposed long time ago, for example by Peter Mitchell (1957 Nature, 180, 134) and reviewed by David (Gadsby DC, 2009, Nat Rev Mol Cell Biol.). This is not the first time that this has been clarified in this paper, and there is no need to emphasize it as a “dual gating model”.

Thank you for your comments. We have changed the wording to “the gating mechanism” in the revised manuscript.

7. Fig.1c

This panel was not found in the previous version. Either bac-mam or PEI transfection using HEK cells are now utilized for many structural and functional analysis of mammalian membrane proteins. For P-type ATPase, bac-mam is first applied for HK-ATPase, and also for many P4- and P5-ATPases. This expression strategy is not new at all, authors just applied this technique for alpha3 isoform.

Thank you for your comments. We have removed this subfigure panel in the revised manuscript.